# Investigating the vertical extent and short-wave radiative effects of ice phase in Arctic summer-time low-level clouds

Emma Järvinen[1], Franziska Nehlert[1], Guanglang Xu[1], Fritz Waitz[1], Guillaume Mioche[2], Regis Dupuy[2], Olivier Jourdan[2], and Martin Schnaiter[1,3]

[1]Karlsruhe Institute of Technology, Karlsruhe, Germany
[2]Laboratoire de Météorologie Physique (LaMP), Université Clermont Auvergne/OPGC/CNRS, Clermont Ferrand, France
[3]schnaiTEC GmbH, Bruchsal, Germany

**Correspondence:** Emma Järvinen (emma.jaervinen@kit.edu)

**Abstract.** Low-level (cloud tops below 2 km) mixed-phase clouds are important in amplifying warming in the Arctic region through positive feedback in cloud fraction, water content and phase. In order to understand the cloud feedbacks in the Arctic region, good knowledge of the vertical distribution of the cloud water content, particle size and phase is required. Here we investigate the vertical extend of the cloud phase and ice phase optical properties in six case studies measured in the European

Arctic during the ACLOUD campaign. Late spring- and summer-time stratiform clouds were sampled in-situ over pack ice, marginal sea ice zone and open ocean surface with cloud top temperatures varying between -15 and -1.5 °C. The results show that, although liquid phase dominates the upper parts of the clouds, ice phase was frequently observed in the lower parts down to cloud top temperatures as warm as -3.8 °C. In the studied vertical cloud profiles, the maximum of average liquid phase microphysical properties, droplet number concentration, effective radius and liquid water content, varied between 23 and 152

$cm^{-3}$, 19 and 26 μm, 0.09 and 0.63 $gm^{-3}$, respectively. The maximum of average ice phase microphysical properties varied between 0.1 and 57 $L^{-1}$ for the ice number concentration, 40 and 70 μm for the effective radius, 0.005 and 0.08 $gm^{-3}$ for the ice water content. The elevated ice crystal number concentrations and ice water paths observed for clouds with cloud top temperatures between -3.8 and -8.7°C can be likely attributed to secondary ice production through rime-splintering. Low asymmetry parameters between 0.69 and 0.76 were measured for the mixed-phase ice crystals with a mean value of 0.72.

The effect of the ice phase optical properties on the radiative transfer calculations was investigated for the four cloud cases potentially affected by secondary ice production. Generally the choice of ice phase optical properties has only a minor effect to the cloud transmissivity and albedo, except in a case where ice phase dominated the upper cloud layer extinction. In this case, cloud albedo at solar wavelengths was increased by 10% when the ice phase was given its measured optical properties instead of treating it as liquid phase. The presented results highlight the importance of accurate vertical information of cloud phase for

radiative transfer and provide a suitable data set for testing microphysical parameterisations in models.

## 1 Introduction

Observations have shown that the Arctic region is particularly sensitive to climate change compared to low latitudes (Rigor et al., 2000; Serreze et al., 2009; Previdi et al., 2021). This sensitivity has been hypothesized to be attributed to myriad of

feedback mechanisms taking place in the region. One important feedback is related to changes in cloud fraction, water content
and phase. Clouds reflect solar radiation and absorb and re-emit thermal longwave radition. In the Arctic region low-level
mixed-phase clouds are abundant (Curry and Ebert, 1992; Morrison et al., 2012; Mioche et al., 2015). Together with stratiform
liquid clouds Arctic mixed-phase clouds have been found be the most important contributors to the Arctic surface radiation
balance by inserting a warming cloud radiative effect in most months except a few months in the summer time when the
short-wave cloud radiative effect overcomes the longwave effect (Shupe and Intrieri, 2004).

In order to correctly simulate cloud-radiation feedbacks in the Arctic a good understanding of the vertical distribution
of the cloud water content, particle size and phase are required (Curry et al., 1996). In-situ observations of cloud vertical
profiles provide a basis for improving this knowledge. Vertical profiles of ice particle microphysical properties in low-level
stratiform clouds over the Arctic Sea have been reported in the European Arctic (Lloyd et al., 2015; Mioche et al., 2017) and
in coastal Alaska (Gultepe et al., 2000; McFarquhar et al., 2007; Jackson et al., 2012). In a study by Mioche et al. (2017),
statistical analysis was performed on cloud vertical profiles combined from four airborne campaigns. The profiles revealed
that supercooled liquid dominates the cloud top, with ice being more prevalent in the lower parts of the cloud without any
significant vertical trend. The average concentration of ice crystals with diameters ($D$) larger than 100 μm was found to be
3 $L^{-1}$ for stratiform clouds with cloud top temperatures between -3 and -25°C. Similar ice phase vertical structures without
clear vertical trends in the microphysical properties were found in the studies of Gultepe et al. (2001), McFarquhar et al.
(2007), and Jackson et al. (2012). Gultepe et al. (2001) summarized ice crystal number concentration observations from two
Arctic campaigns with maximum dimensions greater than 125 μm. The observed average number concentrations, measured
over a wide temperature range from 0 to -45°C, varied between 0.3 and 6.4 $L^{-1}$. McFarquhar et al. (2007) reported average
concentrations of ice crystals with maximum dimensions greater than 53 μm of 2.8±6.9 $L^{-1}$ for stratiform clouds in autumn
with cloud top temperatures varying between -12 and -16°C, while Jackson et al. (2012) reported an average concentration
of ice crystals with maximum dimensions greater than 50 μm of 0.27 $L^{-1}$ for similar cloud top temperatures in April. In
another study by Lloyd et al. (2015), four vertical profiles in spring and summer-time stratocumulus clouds were reported.
Higher median ice crystal concentrations, about 3 $L^{-1}$, were found in summer compared to spring, when the mean ice crystal
concentrations were around 0.5 $L^{-1}$.

Ice crystal concentration above 1 $L^{-1}$ in low-level clouds with cloud top temperatures around -5°C likely cannot be explained
by primary nucleation due to low number of active ice nucleating particles (INPs) in that temperature range (Kanji et al., 2017)
but could be the result of secondary ice production (SIP). For instance, Lloyd et al. (2015) suggested that the summer-time ice
crystal concentrations are the result of rime-splintering and, later, Sotiropoulou et al. (2020) showed in a modelling study that
the observed ice crystal number concentrations can be explained by rime-splintering and collisional break-up. Also Fridlind
et al. (2007) concluded that the ice crystal concentrations observed by McFarquhar et al. (2007) could not be explained by
primary nucleation. Additional evidence from rime-splintering was given by Rangno and Hobbs (2001), who reported ice
crystal concentrations up to 40 $L^{-1}$ in clouds with cloud top temperatures above -10°C in late spring and summer-time Arctic
stratiform clouds. On the contrary, Lawson et al. (2001) reported extremely high ice particle number concentrations (exceeding
1000 $L^{-1}$) for another cloud system measured during the same campaign observed at -12°C, and thus not explainable by

rime-splintering. The most recent evidence of SIP in Arctic low-level clouds was provided by Pasquier et al. (2022), where the authors found that SIP occurred during 40% of the in-cloud measurements in the temperature range from -1 to -24°C performed with a tethered balloon system during the Ny-Ålesund AeroSol Cloud ExperimENT (NASCENT).

Despite the increasing number of in-situ observations in the Arctic region, there is still insufficient understanding of the concentration and vertical profiles of small ice crystals. The vertical distribution of small ice particles (less than 50 µm) is critical to understanding radiative transfer, as well as ice initiation through heterogeneous ice nucleation or through SIP, which are key processes that control the longevity of Arctic mixed-phase clouds (Morrison et al., 2012) and affect precipitation formation (Gultepe et al., 2017).

Previous vertical profiles of low-level Arctic clouds have predominantly been performed over open ocean (Jackson et al., 2012; Lloyd et al., 2015; Mioche et al., 2017) or over coastal areas (Gultepe et al., 2000; Lawson et al., 2001; McFarquhar et al., 2007), and vertical cloud in-situ profiles over Arctic pack ice have not been extensively studied. Therefore, this work aims to increase observational knowledge of the vertical phase composition of Arctic mixed-phase clouds mainly over pack ice by presenting in-situ cloud microphysical observations from the Arctic CLoud Observations Using airborne measurements during polar Day (ACLOUD) campaign, which was conducted northwest of Svalbard (Norway) between May 23 and June 6, 2017.

During ACLOUD, a suite of newer cloud probes was deployed to detect small ice particles down to $D = 9$ µm. We present vertical profiles of liquid and ice phase microphysical and optical properties in six cloud cases, where cloud top temperatures ranged from -15 to -3°C. The vertical information on both liquid and ice phase microphysical properties makes the data set particularly well-suited for testing cloud microphysical parameterizations in models. We also discuss the vertical variability of liquid and ice phase optical properties and the implications of ice phase for the radiative properties of low-level clouds.

## 2 Details of the field experiment

### 2.1 Meteorological situation

The ACLOUD aircraft campaign performed 22 research flights between 23 May and 26 June 2017 from Svalbard towards the Arctic ocean. The synoptic development during ACLOUD can be separated into three periods (Knudsen et al., 2018). During the first days of the campaign a seasonally unusual cold air outbreak brought cold and dry Arctic air from the north. This cold period (23 to 29 May) was followed by a warm period (30 May to 12 June) with warm and moist air intrusions into the region caused by a strong southwesterly flow due to a high pressure system located over the Greenland Sea. During 11 and 12 June, northerly winds started to dominate the lower troposphere, indicating the end of the moist air intrusion and the beginning of a normal period (13 to 26 June) where both the temperature and moisture were close to long-term averages recorded in Ny-Ålesund.

## 2.2 Instrumentation

The Polar 6 was equipped with in-situ instrumentation for the characterisation of cloud hydrometeors, aerosol particles and trace gases. The cloud instrumentation included the following three instruments used here:

- The Small Ice Detector Mark 3 (SID-3; Hirst et al. (2001)), which detects individual cloud particles passing a 532 nm laser beam using two nested trigger detectors with a half angle of $9.25°$ symmetrically located at $50°$ relative to the forward direction. The trigger signal is recorded as a histogram with a maximum rate of 11 kHz that can be used to derive

particle size distributions by using the procedure described in Vochezer et al. (2016). For a sub-set of triggered particles a two-dimensional (2-D) scattering pattern is recorded that can be analysed for particle sphericity by a specifically developed image analysis software (Vochezer et al., 2016). Occasionally, coincidence sampling in the camera field of view causes optical distortions of the 2-D scattering patterns of liquid droplets and, consequently, a misclassification of such scattering patterns to be aspherical by the classification software. For the subsequent identification and re-

classification of coincidence scattering patterns a machine learning (ML) algorithm was developed (see Appendix A for details). From the numbers of observed spherical and aspherical 2-D scattering patterns the fractions of spherical and aspherical particles are derived. Multiplication of those number-based fractions with the total particle size distribution yields phase-specific particle size distribution. The uncertainty due to the fact that the imaged particles are a subset of all sampled particles can be estimated from the Clopper–Pearson confidence limits as discussed in Vochezer et al. (2016).

These phase-specific particle size distribution are issued in the ACLOUD data set (Schnaiter and Järvinen, 2019a).

- The Particle Habit Imaging and Polar Scattering (PHIPS) probe, which is a combination of a polar nephelometer and a high-resolution cloud particle stereo-imager (Abdelmonem et al., 2016; Schnaiter et al., 2018; Waitz et al., 2021). The two parts of the instrument are combined by a trigger detector so that both imaging and scattering measurements are performed on the same single particle. The polar nephelometer has 20 channels ranging from 18 to $170°$, with an angular

resolution of $8°$. The measured single-particle light scattering functions (Schnaiter and Järvinen, 2019c) can be used to derive particle sphericity and size distributions of spherical and aspherical particles using the methods discussed in Waitz et al. (2021). Here particle size distributions were calculated for 10-s time resolution corresponding to a lower detection limit of about $2~\mathrm{L}^{-1}$. The uncertainties in the number concentrations of droplets and ice particles are 20% and 40%, respectively (Järvinen et al., 2022). The stereo-microscopic imager consists of two camera and microscope assemblies

with an angular viewing distance of $120°$ acquiring bright field stereo-microscopic images of individual cloud particles. During ACLOUD, two different magnifications of 6x and 8x were set for the two PHIPS microscopes of camera 1 and 2 corresponding to optical resolutions of $\sim3.5$ μm and $\sim2.3$ μm, respectively.

- The Cloud Imaging Probe (CIP; Baumgardner et al. (2001)), which uses a linear-array technique to acquire two-dimensional shadow images of cloud particles. The CIP has a nominal size range from 25 to 1550 μm with 25 μm

pixel-resolution. Ice phase cloud particles are separated from liquid phase particles following the approach of Crosier et al. (2011) based on a circularity parameter (circularity larger than 1.25 and image area larger than 16 pixels). Only

these non-spherical particles were used to calculate ice-phase properties. The ice particle size distributions were calculated for non-spherical particles using area-equivalent diameter. Possible contamination by shattering artifacts were removed using inter-arrival time analysis and image processing according to Field et al. (2006). The remaining combined uncertainty in the number concentration is 50% (Baumgardner et al., 2017). The resolution of the CIP data products is given in 1-Hz (Dupuy et al., 2019) and here we averaged the data over 10-s periods to increase the counting statistics corresponding a lower detection limit of $0.01 \, \mathrm{L}^{-1}$.

On Polar P6 high-frequency measurements of wind vector and air temperature were performed in a nose boom using an Aventech five-hole probe and an open-wire Pt100 installed sidewards in a Rosemount housing (Hartmann et al., 2018). Humidity was measured with 1-Hz resolution with a Vaisala HMT-333, which includes a temperature and HUMICAP humidity sensor. Based on the temperature measurements (uncertainty of 0.1 K), the humidity data were corrected for adiabatic heating (Hartmann et al., 2018). In this paper we use merged thermodynamic measurements providing aircraft position, air pressure, temperature, relative humidity, and the horizontal wind vector at a resolution of 1-Hz (Hartmann and Chechin, 2019). Aerosol particle concentration in the nominal size range from 60 to 1000 nm were measured with the ultra-high sensitivity aerosol spectrometer (UHSAS, Cai et al. (2008)) that was installed behind the counter-flow virtual impactor (CVI) (Mertes, 2019).

## 2.3 Calculation of microphysical and optical parameters

### 2.3.1 Concentrations of spherical and aspherical particles

Total concentration of spherical particles in the size range from 5 to 700 μm was calculated by combining the total concentration of spherical particles measured by SID-3 (between 5 and 42 μm) and PHIPS (between 60 and 700 μm). No extrapolation was performed to cover the missing size range between 42 and 60 μm as the concentration and LWCs of spherical particles measured by PHIPS were typically more than two magnitudes lower than spherical particles measured by SID-3.

Total concentration of aspherical particles in the size range from 9 to 1550 μm was calculated by combining the total concentration of aspherical particles measured by SID-3 in the size range from 9 to 30 μm, the total concentration of aspherical particles measured by PHIPS in the size range from 30 to 200 μm and the total concentration of aspherical particles measured by the CIP in the size range from 200 to 1550 μm. The size limits were chosen to maximise the counting statistics and optimise phase discrimination certainty. Occasionally, indications of shattering were seen in the PHIPS data and sometimes also in the SID-3 data (see supplementary information). If shattering was observed, the PHIPS and SID-3 data were removed from analysis so that the total concentrations were only given for particles >200 μm measured by CIP.

### 2.3.2 LWC, IWC, LWP and IWP

The LWC (IWC) for each cloud microphysical probe was calculated using the following equation

$$LWC(IWC) = \sum_{D_{min}}^{D_{max}} n(D)M(D), \tag{1}$$

where $n(D)$ is the number of spherical (aspherical) particles in a size bin and $M(D)$ is the mass of a particle having a diameter corresponding to the bin mean diameter.

For derivation of LWC, $M(D)$ was calculated for spherical particles with a density of $1\,\mathrm{g\,cm^{-3}}$. Since light scattering instruments typically have a systematic measurement uncertainties between 10 and 30% in concentration, we consider the LWC to have a systematic uncertainty up to 30%.

For derivation of IWC from SID-3 measurements $M(D)$ was calculated assuming spherical particles with a density of 0.91 $\mathrm{g\,cm^{-3}}$. IWC was calculated from PHIPS and CIP measurements using mass-dimensional (M-D) relations. Since there are several M-D relations in the literature depending of the cloud type and mixture of habits we performed sensitivity studies using M-D relations from Brown and Francis (1995) (hereafter BF95), McFarquhar et al. (2007) and habit-dependent M-D relationships measured by Mitchell et al. (1990) and revised by Lawson and Baker (2006). Based on the habits observed by the PHIPS, the following habit-dependent M-D relations were chosen: needles, rimed needles, hexagonal plates and mixture of all habits. The highest IWC was retrieved using the BF95 M-D relation and 40 to 60% lower IWC was retrieved using M-D relations by McFarquhar et al. (2007) and habit mixtures of all habits and plates by Lawson and Baker (2006) (Fig. S18). The lowest IWCs (by 85% compared to Brown and Francis (1995)) was retrieved for needles and rimed needles.

The integrated liquid and ice water paths were calculated from in-situ measurements according to the following equation

$$WP = \int\limits_{ground}^{cloudtop} WC z \, dz, \tag{2}$$

where $WP$ is either the liquid water path (LWP) or ice water path (IWP), $WC$ either LWC or IWC and $z$ is the height between ground and cloud top. For calculation of IWP we used the IWC calculated using Brown and Francis (1995) M-D relation. The 10-s LWC and IWC values were first binned to height bins of 10 m and averaged before performing the numerical integration.

### 2.3.3 Extinction coefficient and effective radius

The extinction coefficient for visible wavelengths for liquid and ice phase was calculated using the following equation

$$\beta_{ext} = \sum_{D_{min}}^{D_{max}} n(D)\sigma_{ext}(D), \tag{3}$$

where $\sigma_{ext}(D)$ is the extinction cross section of a particle having a diameter corresponding to the bin mean diameter.

For spherical particles in the SID-3 size range, the $\sigma_{ext}(D)$ was calculated by multiplying the geometrical cross section with the extinction efficiency ($Q_{ext}$) calculated using the Mie theory for $532\,\mathrm{nm}$. For PHIPS and CIP size range geometrical optics assumption was used, where the scattering cross section is two times the geometrical cross section.

For calculating the effective radius ($r_{eff}$) several definitions are available in the literature (McFarquhar and Heymsfield, 1998). Here the following definitions were used to calculate the effective radius of the liquid ($r_{e,w}$) and ice phase ($r_{e,i}$)

$$r_{e,w} = \frac{3LWC}{2\rho_w\beta_{ext,w}} \tag{4}$$

$$r_{e,i} = \frac{3IWC}{2\rho_i\beta_{ext,i}}, \tag{5}$$

where $\rho_{w,i}$ is the bulk density of water or ice. For calculation of $r_{e,i}$ we used the IWC calculated using BF95 M-D relation.

### 2.3.4 Ice crystal asymmetry parameter and complexity parameter

The asymmetry parameter ($g$) of ice crystal ensembles was retrieved from the average ice crystal angular scattering functions following the method presented in Xu et al. (2022). The retrieval is based on the assumption that in geometrical optics range, the following relation can be used to derive asymmetry parameter,

$$g = \frac{1}{2\omega_0}[(2\omega_0 - 1)g_{GO} + g_D], \tag{6}$$

where $g_{GO}$ and $g_D$ are the asymmetry parameter contributed by geometric-optics and diffraction, respectively. As the diffraction phase function is highly peaked, $g_D$ is very close to unity. According to the analysis of scalar diffraction theory, most of the diffracted energy will be confined into the angular range of $\theta < 7/x$ (in radian), where $x$ is the size parameter. On a logarithmic scale, $g_D(d)$ can be approximated by a polynomial of degree 4, i.e,

$$\begin{aligned}
g_D(d) = &- 5.9270 \times 10^{-5} - 0.00130 \times ln(d) - 0.01087 \times (ln(d))^2 \\
&+ 0.04093 \times (ln(d))^3 + 0.94029 \times (ln(d))^4,
\end{aligned} \tag{7}$$

where $d$ is the particle diameter. The geometric-optics contribution $g_{GO}$ can be obtained from polar nephelometer measurements by extrapolating the measurements by expanding the measured function in terms of series of Legendre polynomials. The asymmetry parameter is then the first moment of scattering phase function with respect to Legendre polynomial. Additionally, Xu et al. (2022) defined that the decay of the expansion coefficients gives information on the smoothness and isotropic degree of the phase function and defined a so-called $C_p$ parameter,

$$C_p = (\sum_{l=0}^{\infty} |\hat{c}_{GO,l}|)^{-1}, \tag{8}$$

where $\hat{c}_{GO,l}$ is the expansion coefficients of phase function due to the reflection-refraction of light ray by using a series of Legendre polynomials. Here we calculated one $g$ and $C_p$ value for each horizontal leg if a minimum of 20 ice particles were observed by PHIPS.

### 2.4 Habit classification

Ice crystal habits were manually classified using a classification scheme following Bailey and Hallett (2009). In our manual habit classification we apply a tree-based classification scheme. On the main node we distinguish between single and poly-crystals. The next level is the subdivision into plate-like, column-like and mixed growth. The leaf nodes are the habits, namely

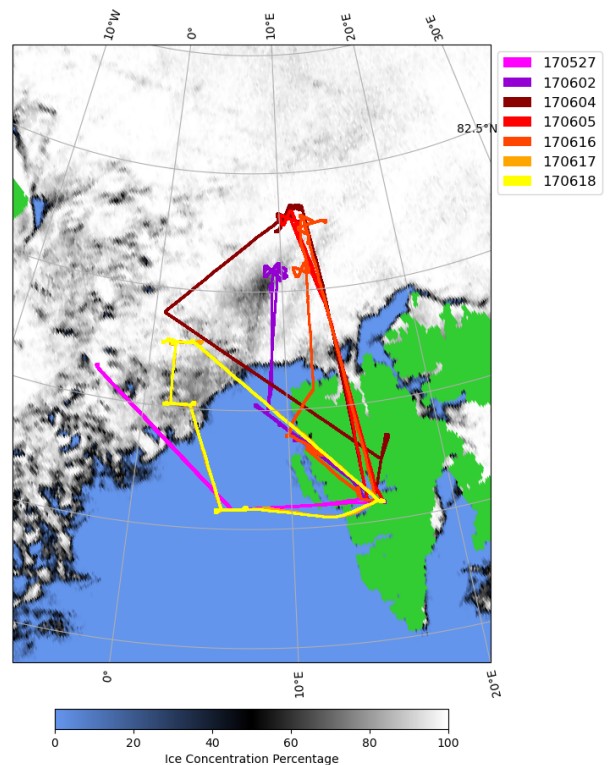

**Figure 1.** Flight paths for the ACLOUD flights with Polar 6 aircraft included in the analysis overlaid on pack ice extend as of 2 June. pack ice data is derived from measurements of the Advanced Microwave Scanning Radiometer 2 (AMSR2) at 89 GHz (www.seaice.uni-bremen.de; Spreen et al. (2008)).

plate, sectored plate, skeletal plate, dendrite, column, needle, bullet, side plane, bullet rosette, capped column, capped bullet rosette and mixed rosette. A more in depth description of our classification scheme can be found in the supplementary information. Depending on the image quality, crystal size and orientation, classification is done to the level where it can clearly be determined. In addition to its habit, attributes like riming and aggregation, were assigned to each ice crystal.

## 3 Vertical profiles of cloud microphysical properties

Low-level clouds (cloud tops below 2 km) were observed during all measurement days of the ACLOUD campaign in a sector northwest of Longyearbyen (as shown in Fig. 1). Space-borne remote sensing observations indicate that cloud top heights were lower during the warm period compared to the cold period, whereas during the normal period, higher and more variable cloud tops were observed (refer to Fig. 3 in Wendisch et al. (2019)). In this study, we discuss the detailed vertical profiles of cloud microphysical and optical properties that were measured during these three distinct meteorological regimes.

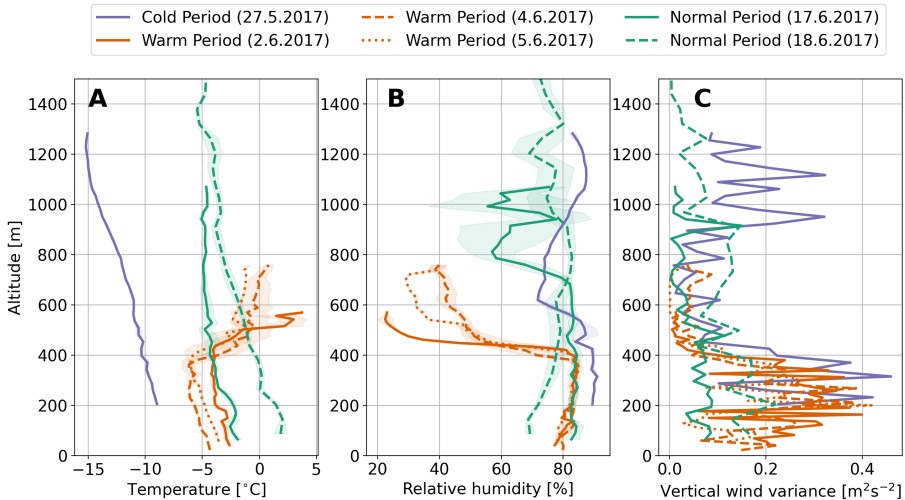

**Figure 2.** Average temperature (A), relative humidity with respect to water (B) and vertical wind variance (C) for the vertical profiles performed during cold period (blue), warm period (red) and normal period (green). Note that the lines represent the average values during the entire cloud sampling period. The shaded area in A and B shows the standard deviation. Note that the absolute value of relative humidity is not considered to be reliable and the values in panel B should only be considered to represent the trend in the relative humidity.

**Table 1.** Vertical cloud profiles included in the study. For each profile the number of horizontal sampling legs, the cloud base, the cloud top, the cloud top temperature (T), liquid water path (LWP), ice water path (IWP) and aerosol number concentration ($N_a$) for aerosol particles with D >60 nm is given. If cloud base and cloud top were crossed multiple times the range for cloud base and cloud top is given.

| Date | Number of horizontal sampling legs | Cloud base m a.s.l. | Cloud top m a.s.l. | Cloud top T °C | LWP $g\,m^{-2}$ | IWP $g\,m^{-2}$ | $N_a$ above $cm^{-3}$ |
|---|---|---|---|---|---|---|---|
| Cold Period | | | | | | | |
| 27 May | 3 | 230 | 1300 | -15.2 | 40.3 | 0.9 | 73 |
| Warm Period | | | | | | | |
| 2 June | 3 | 189 (177-201) | 440 | -4.6 | 82.5 | 9.5 | 175 |
| 4 June | 5 | 98 (93-103) | 433 (374-433) | -6.7 | 57.5 | 4.1 | 126 |
| 5 June | 4 | 206 | 435 (425-445) | -6.5 | 48.0 | 6.8 | 134 |
| Normal Period | | | | | | | |
| 17 June | 4 | 73-105 | 473-934 | -5.2 | 31.4 | 8.8 | 162* |
| 18 June | 3 | 441-824 | 1225-1320 | -5.3 | 22.7 | 43.7 | - |

*below cloud value

We collected samples of the low-level clouds at a single geographical location by flying in either a double-triangle pattern or stacked horizontal legs, as shown in Figure 1. The only exception was on 27 May, during the cold period, when we took samples along a horizontal transect over the marginal sea ice zone. During in-cloud sampling on each horizontal leg, we collected data for 7 to 10 minutes, which was enough to derive cloud microphysical properties from single-particle spectrometers. To generate the vertical profiles, we divided the 10-second measurement data into equidistant altitude bins and calculated statistical properties such as mean and standard deviation for liquid and ice phase microphysical properties. More information on how we generated the vertical profiles from the 10-second data can be found in the supplementary information.

For the warm period, we present vertical profiles as a function of normalised cloud altitude, $Z_n$, which is defined as follows (Mioche et al., 2017):

$$Z_n = \frac{Z - Z_b}{Z_t - Z_b} \quad \text{for } Z_b < Z < Z_t$$

$$Z_n = \frac{Z}{Z_b} - 1 \quad \text{for } Z < Z_b$$

where $Z$ is the altitude corresponding aircraft measurements, and $Z_t$ and $Z_b$ the cloud liquid layer top and base, respectively. A threshold LWC of 0.01 $\mathrm{g\,m^{-3}}$ was used to define the liquid layer top and base. A detailed list of the vertical cloud profiles can be found in Table 1.

### 3.1 Vertical profile over marginal sea ice zone during cold period on 27 May

On 27 May a research flight was performed off the west coat of Svalbard, where stratus clouds were sampled over the marginal sea ice zone between 79.8 and 78.6°N. Figure 2 shows the vertical profile of temperature during the period of cloud sampling. No clear temperature inversion was observed on this day. The sampled cloud system was multi-layered consisting of cloud layer between 1080 and 1317 m with a cloud top temperature of -15.2°C and a lower cloud layer ranging between 186 and 630 m with a cloud top temperature of -11°C. Precipitation was observed between the cloud layers.

The multi-layered cloud system was sampled in one ascent with three straight legs performed in the lower cloud at altitudes of 490, 410 and 360 m. The mean vertical profiles of liquid cloud properties are shown in Figure 3. Mean cloud droplet number concentration up to 23 $\mathrm{cm^{-3}}$ were observed in the upper cloud layer and somewhat higher mean droplet concentrations up to 30 $\mathrm{cm^{-3}}$ were observed in the lower cloud layer. The effective diameter was observed to increase with altitude with larger mean effective diameters up to 26 µm observed in the lower cloud layer compared to the upper cloud layer where the mean effective diameters up to 19 µm were observed. A few large (D>60 µm) drizzle droplets were observed in the PHIPS images in the cloud layers. LWC mean values of 0.1 and 0.2 $\mathrm{g\,m^{-3}}$ and extinction coefficient of 18 and 23 $\mathrm{km^{-1}}$ were observed in the upper and lower cloud, respectively. However, it should be noted that the lower cloud layer was sampled closer to the pack ice edge compared to the upper cloud layer, which might explain some of the differences seen in the microphysical properties.

During the cold period case study, the liquid cloud properties observed were comparable to those reported in McFarquhar et al. (2007) for clouds with similar cloud top temperatures. However, the observed droplet concentrations were lower by a factor of 2 to 4 compared to other Arctic cloud situations influenced by cold air outbreaks reported in Lawson et al. (2001),

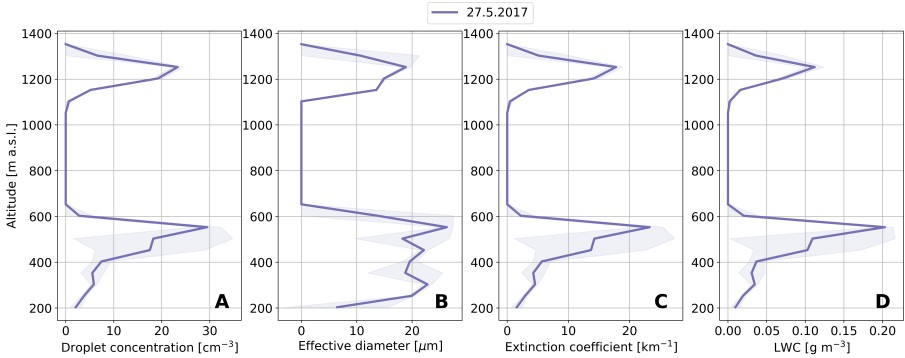

**Figure 3.** Vertical profile of average droplet concentration (A), droplet effective diameter (B), droplet extinction coefficient (C) and liquid water content (LWC) (D) performed on 27 May during the cold period. The 10-s average measurement data was binned in altitude bins with bin width of 49.3 m for calculating the statistics. The shaded area illustrates the standard deviations. The statistical uncertainty in droplet number concentration was below $0.3 \ \mathrm{cm}^{-3}$. Cloud sampling was performed between 78.7°N and 79.8°N and 3.3°W and 4.1°E.

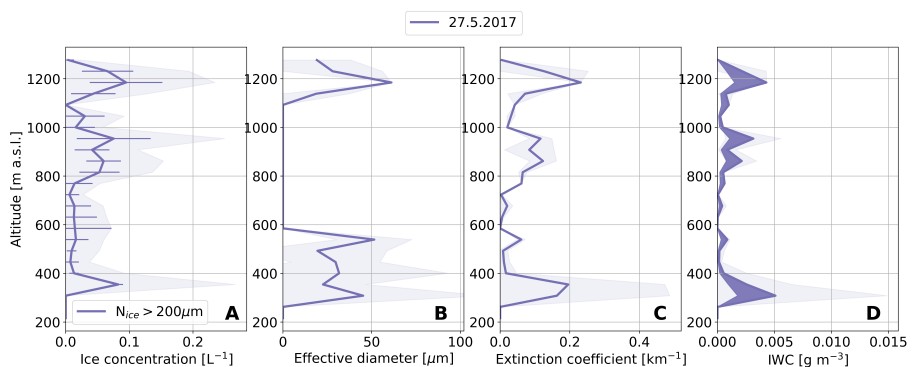

**Figure 4.** Vertical profile of average concentration of ice particles with D >200 μm (A), ice particle effective diameter (B), ice particle extinction coefficient (C) and ice water content (IWC) (D) performed on 27 May during the cold period. The 10-s average measurement data was binned in altitude bins with bin width of 49.3 m for calculating the statistics. The shaded area illustrates the standard deviations. In panel A the horizontal error bars the statistical uncertainty in CIP ice number concentrations calculated using $\sigma(N_{ice})/\sqrt{n}$, where $n$ is the number of counts used to calculate $N_{ice}$ >200 μm. In panel D the dark shaded area illustrates the range in IWC when using M-D relations of BF95 (upper limit) and McFarquhar et al. (2007) (lower limit). Note that ice particle microphysical properties were retrieved using only CIP due to observed shattering in PHIPS and SID-3 probes.

Jackson et al. (2012), and Mioche et al. (2017). It is worth noting that these previous measurements were conducted over open water, which could partly explain the differences. For instance, Mioche et al. (2017) observed similar LWC in cold air outbreak-influenced clouds, but these clouds had effective droplet diameters around 15 μm.

Figure 4 shows the vertical profiles of ice phase microphysical properties. Only CIP was used to retrieve ice phase properties for ice particles larger than 200 μm in diameter due to indications of shattering observed in the PHIPS stereo-images. For

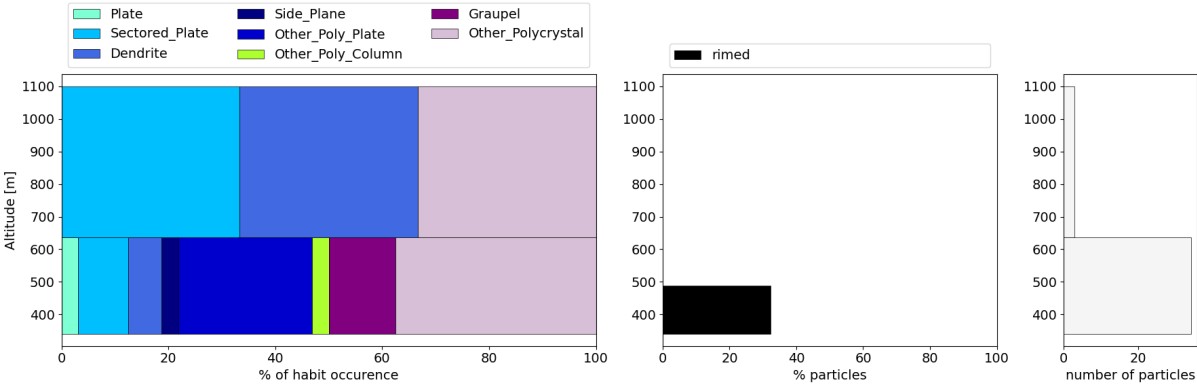

**Figure 5.** Analysis of ice crystal habits during cloud sampling on 27 May. The bar between 1317 and 630 m represents the habits in the upper cloud and in the precipitation zone. The lower bar represent the habits in the lower cloud.

precaution, also SID-3 ice data was excluded from the analysis. From those PHIPS stereo-images that were not influenced by shattering it can be seen that the dominant habits in the multilayered cloud system were dendrites, sectored plates and other polycrystaline plates (Fig. 5) in accordance with laboratory studies of Bailey and Hallett (2009). Riming was also present in the lower cloud layer. On average, the concentration of ice particles with diameters larger than 200 μm was below 0.1 L$^{-1}$. This value is similar to the expected ice-nucleating particle (INP) concentrations for cloud top temperatures of -15°C (Kanji et al., 2017), and falls within the range of INPs observed by Li et al. (2022) in Ny Ålesund. Thus, primary ice nucleation could account for the observed ice particle number concentrations. The low ice crystal concentrations translate to low IWC of 0.005 g m$^{-3}$ or below, and extinction coefficient below 0.23 km$^{-1}$. Ice crystal effective diameter was around 60 μm in the upper cloud and 40 μm in the lower cloud.

## 3.2 Vertical profiles over pack ice during warm period on 2, 4 and 5 June

On 2 June the moist and warm air intrusion caused development of cloudiness to the Svalbard area and a fairly uniform low-level cloud deck was observed starting from Ny Ålesund reaching to Polarstern pack ice camp. On the following days the uniform cloud deck persisted. On 2, 4 and 5 June three vertical cloud profiles were performed near Polarstern sampling the low-level cloud deck over pack ice (Fig. 1). Figure 2 shows that on those three days similar vertical profiles of temperature were observed with cloud top temperatures between -4.6 and -6.7°C capped by a strong temperature inversion of about 8°C. Inversion is a dominant feature in the Arctic environment, particularly during the coldest half of the year (e.g., Kahl, 1990; Kahl et al., 1992; Serreze et al., 1992). On all three days a single layer cloud was observed with occasional visible cirrus clouds in the horizon. Cloud tops were observed to be around 400 m and cloud thicknesses were around 300 m.

Figure 6 displays the mean vertical profiles of liquid cloud properties observed during the three days of stratiform cloud sampling near Polarstern over pack ice. All profiles exhibit an increase in droplet concentration, effective droplet diameter, extinction coefficient, and LWC with altitude, followed by a subsequent decrease as cloud top is approached. The highest

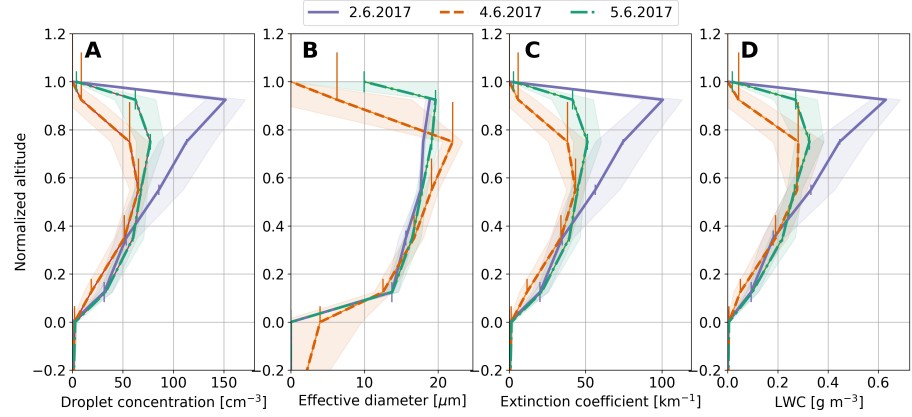

**Figure 6.** Vertical profiles of average droplet concentration (A), droplet effective diameter (B), droplet extinction coefficient (C) and liquid water content (LWC) (D) measured on 2, 4 and 5 June during the warm period. The shaded area illustrates the standard deviations. The mean and standard deviation were calculated based on 10-s aircraft observations for normalised altitude bins having bin edges at -0.6, -0.3, 0, 0.25, 0.45, 0.65, 0.85, 1, 1.05. The vertical error bars indicate the uncertainty in normalised altitude. The statistical error in droplet number concentrations were below $0.1 \, \mathrm{cm^{-3}}$. Cloud sampling was performed between 81.1°N and 82.1°N and 8.5°E and 11.7°E.

LWC values were found on 2 June, between $Z_n$ =0.85 and 1, while on 4 June and 5 June, the maximum LWC occurred around $Z_n$ =0.5 and $Z_n$ =0.7, respectively. The maximum mean droplet number concentrations were $152 \, \mathrm{cm^{-3}}$ on 2 June and decreased to 66 (78) $\mathrm{cm^{-3}}$ on 4 June (5 June). This decline in droplet concentration correlated with the decrease in aerosol number concentration above the clouds, which fell from $173 \, \mathrm{cm^{-3}}$ to 126 (134) $\mathrm{cm^{-3}}$.

Effective droplet diameters remained around 20 µm near the top of the clouds on all three days. On 2 June, the extinction
coefficient and LWC were the highest, at $101 \, \mathrm{km^{-1}}$ and $0.63 \, \mathrm{g \, m^{-3}}$, respectively, while on 4 June and 5 June, they were lower, at 44 (50) $\mathrm{km^{-1}}$ and 0.29 (0.33) $\mathrm{g \, m^{-3}}$, respectively. The warm period clouds contained 2 to 4 times more droplets and had a larger LWC by a factor of 1.5 to 3 than the cold period clouds, which can be attributed to the intrusion of moister air with a higher aerosol loading.

Comparison with theoretical adiabatic LWC profiles (not shown here) revealed that the clouds measured during the warm
period were superadiabatic, possibly due to stronger updrafts, entrainment of humid air from above, or radiative cooling at cloud top. The LWC values over pack ice were similar to those observed in similar warm air intrusion situations over open sea surfaces by Mioche et al. (2017), but 3 times lower than the LWC in clouds measured over open sea surfaces during ACLOUD (Dupuy et al., 2018).

Figure 7 displays the mean vertical profiles of ice phase microphysical properties. The concentration of small ice was only
able to be derived for the upper half of the cloud due to shattering that was observed in the lower parts. In the upper half of the cloud all profiles exhibit a slight increase of IWC and extinction coefficient with decreasing altitude. However, definite conclusions about the IWC vertical trends cannot be reached due to uncertainties linked to the M-D relationship used to calculate IWC. A small amount of ice was detected at the cloud tops, indicating that the top layer of the cloud is comprised almost

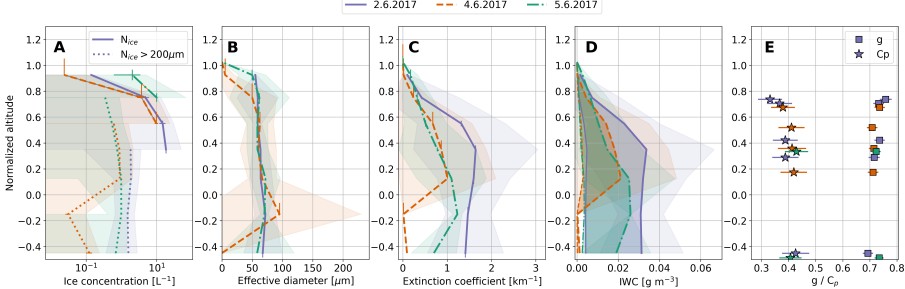

**Figure 7.** Vertical profiles of average ice particle concentration ($N_{ice}$) (A), ice particle effective diameter (B), ice particle extinction coefficient (C) and ice water content (IWC) (D) performed on 2, 4 and 5 June during the warm period. The shaded area illustrates the standard deviations. The mean and standard deviation were calculated similar to liquid phase. The vertical error bars indicate the uncertainty in normalised altitude and the horizontal error bars the statistical uncertainty in $N_{ice}$. For PHIPS and CIP the statistical uncertainty was calculated using $\sigma(N_{ice})/\sqrt{n}$, where $n$ is the number of counts included in calculation of the mean concentration. For SID-3 the statistical uncertainty is calculated using Clopper–Pearson confidence limits. In panel D the dark shaded area illustrates the range in IWC when using M-D relations of BF95 (thicker lines) and a habit dependent M-D relation of Lawson and Baker (2006) for needles (thinner lines). Note that ice particle microphysical properties in the lower parts of the cloud were retrieved using only CIP for ice crystals D > 200 μm ($N_{ice>200\mu m}$) due to observed shattering in PHIPS and SID-3 probes. Panel E shows the asymmetry parameter ($g$) and complexity parameter ($C_p$) derived from PHIPS. One value was derived for each straight leg within the cloud. The horizontal error bars in panel E illustrate the retrieval uncertainty.

entirely of supercooled liquid droplets. The maximum mean ice crystal number concentrations, during periods unaffected by
shattering, varied between 10 and 18 L$^{-1}$. Effective diameters ranged from 50 to 70 μm near the cloud base and decreased towards the cloud top. Extinction coefficients ranged from 1 to 1.7 km$^{-1}$. Mean IWC values peaked between 0.02 and 0.034 g m$^{-3}$ in the lower half of the cloud when assuming the BF95 M-D relationship. Given that the cloud was mostly composed of unrimed needles (as shown in Fig. 8), we also calculated IWC using a habit-dependent M-D relationship for needles by Lawson and Baker (2006), resulting in significantly lower IWCs by an order of magnitude. While this wide range highlights
the uncertainty of IWC measurements when using M-D relationships, it should be noted that the lower limit for IWC is likely unrealistic due to the fact that the cloud was not entirely composed of one particle habit.

Upon investigating particle habits, it was determined that the most prevalent ice habits within the PHIPS measurement range (D<1 mm) were single crystals, taking on the form of needles (31.0 %) and columns (16.4 %) (as depicted in Fig. 8). This finding stands in contrast to earlier studies, which identified irregular (polycrystal) habits as being the dominant types in
Arctic mixed-phase clouds (Korolev et al., 1999; Mioche et al., 2017). Additionally, a significant proportion of the ice crystals observed (38.5 %) displayed riming. Furthermore, the ice crystal habits exhibited a vertical trend, whereby the fraction of single columns increased towards the cloud top. Although an increase in rimed particle fraction was observed at greater cloud heights during all days, no definitive conclusions regarding the vertical structure in riming could be drawn due to low statistics.

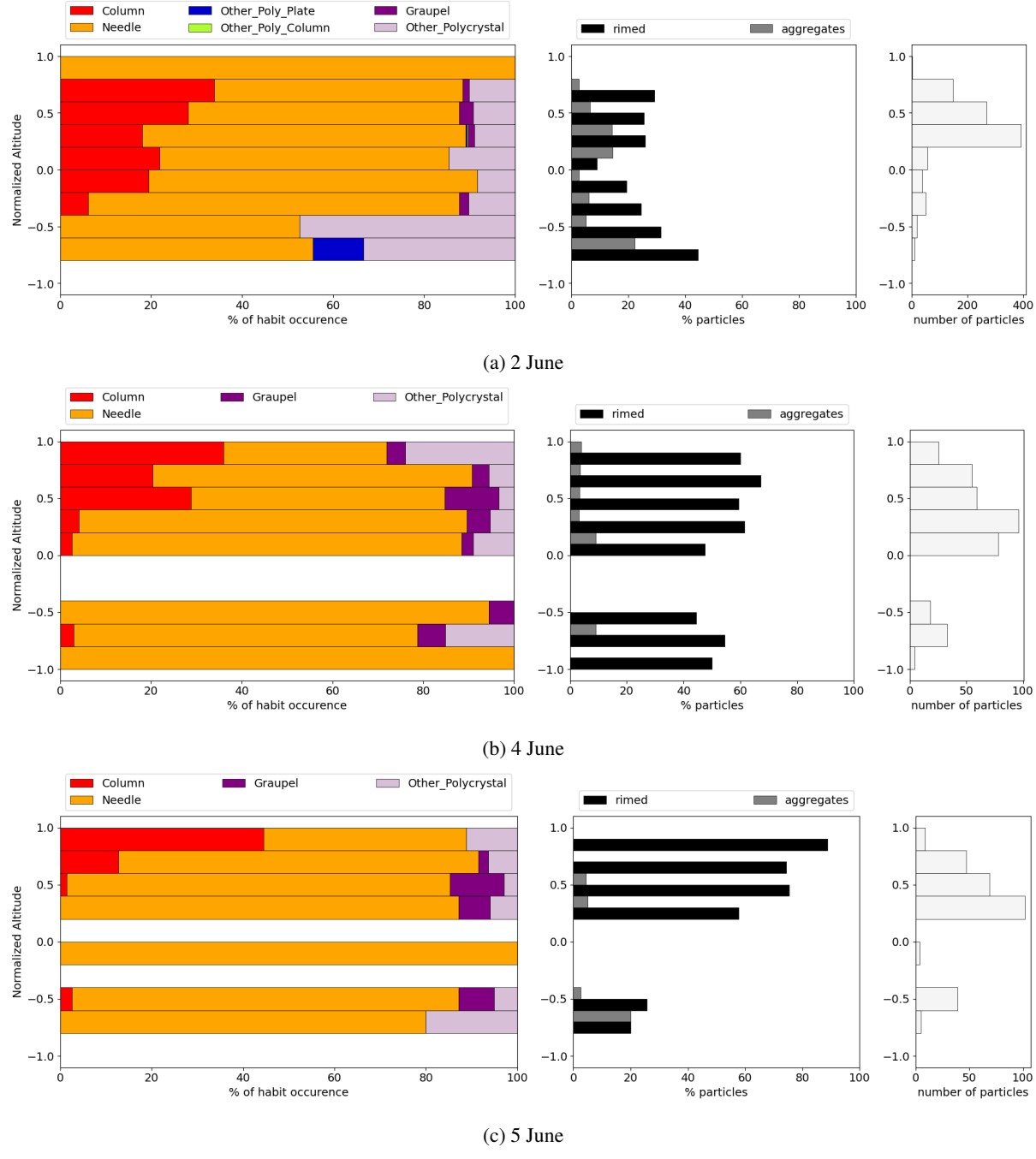

**Figure 8.** Analysis of ice crystal habits imaged by the PHIPS during cloud sampling on 2, 4 and 5 June during warm period.

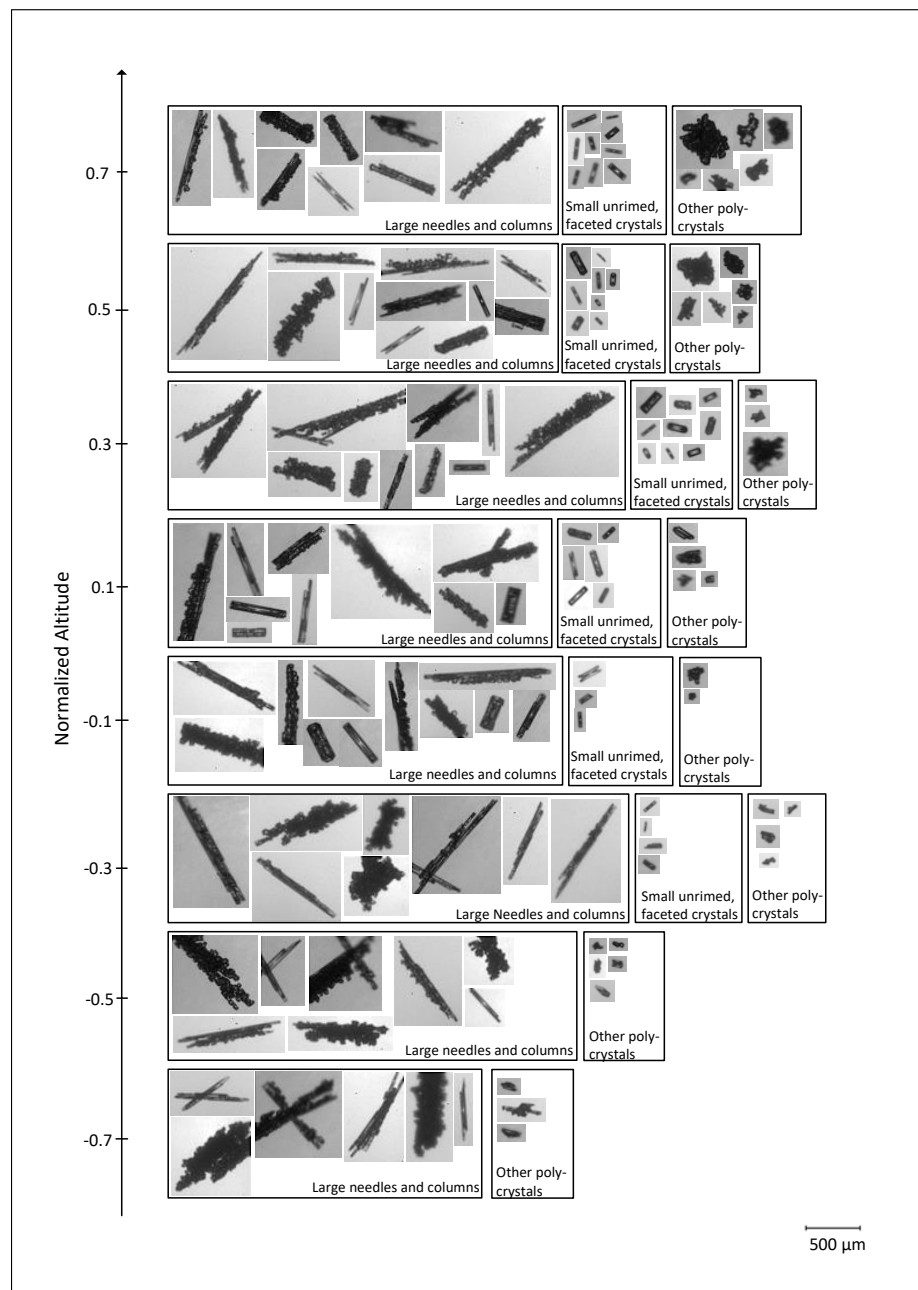

**Figure 9.** Examples of ice crystals representing three categories that were frequently observed on 2 June: large (D>500 μm) needles and columns, smaller unrimed faceted crystals and other polycrystals. Images are from the PHIPS probe.

### 3.3 Vertical profiles during normal period on 17 and 18 June

In the final two weeks of the ACLOUD campaign, adiabatically warmed air from the west and north dominated the study region (Knudsen et al., 2018) leading to more variable cloud properties (Wendisch et al., 2019). On 16 June northerly flows brought

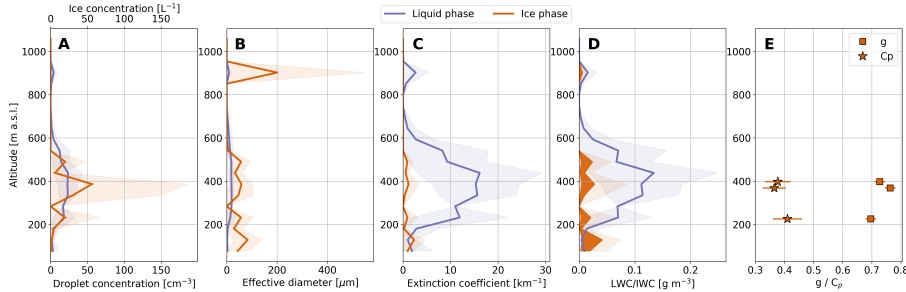

**Figure 10.** Vertical profile of average liquid phase (purple) and ice phase (red) number concentration (A), effective diameter (B), extinction coefficient (C) and condensed water content (LWC/IWC) (D) performed on 17 June during the normal period. The shaded area illustrates the standard deviations. The horizontal error bars in panel A show the statistical uncertainty in $N_{ice}$. In panel D the dark shaded area illustrates the range in IWC when using M-D relations of BF95 (upper limit) and a habit dependent M-D relation of Lawson and Baker (2006) for needles (lower limit). The 10-s average measurement data was binned in altitude bins with bin width of 52 m for calculating the statistics. Cloud sampling was performed between 80.0°N and 80.4°N and 1.4°E and 5.8°E.

colder air from north and west of Svalbard to the study region generating a solid cloud deck from Svalbard to Polarstern vessel. On the following days the cloud deck persisted but became spatially more inhomogeneous often with multi-layered structure. On 17 June cloud profiling was performed over pack ice near Polarstern and on 18 June over open sea surface (Fig. 1). The
cloud top temperatures were observed around -5°C without a temperature inversion (Fig. 2).

The mean vertical profiles of liquid and ice phase microphysical properties measured over pack ice on 17 June are shown in Figure 10. On that day, the cloud system near Polarstern consisted of a mixed lower layer topped with a thin stratus layer, with the cloud base observed to vary between 73 and 105 m. Precipitation was observed below the cloud. The cloud droplet number concentrations ranged from around 4 cm$^{-3}$ in the upper stratus layer to 24 cm$^{-3}$ in the lower layer, with an effective
diameter of approximately 20 µm. The liquid extinction coefficient and LWC reached maximum mean values of 16 km$^{-1}$ and 0.13 g m$^{-3}$, respectively.

Ice crystals were observed throughout the cloud system, and around 400 m, the average ice crystal number concentrations were up to 57 L$^{-1}$, which were dominated by the number of small (D<200 µm) ice crystals. The ice crystal effective diameter was around 50 µm in the lower, thicker cloud and up to 200 µm in the upper stratus layer. The ice phase extinction coefficient
and IWC within the cloud were significantly lower than those for the liquid phase, around 1 km$^{-1}$ and below 0.02 g m$^{-3}$, respectively. The dominant ice crystal habits observed were columns and needles with rimed fractions between 30 and 40% (Fig. 11a).

On 18 June a vertical profile was performed over opean ocean. The measured stratocumulus layer differed significantly from the clouds measured over pack ice: strong convective clusters penetrating through the stratocumulus deck from below were
observed. The convective clusters were separated from each other by distances of order of 10 km. The cloud base was observed to be variable along the horizontal sampling leg rising towards the turning point. The cloud base was penetrated at 441 and 824

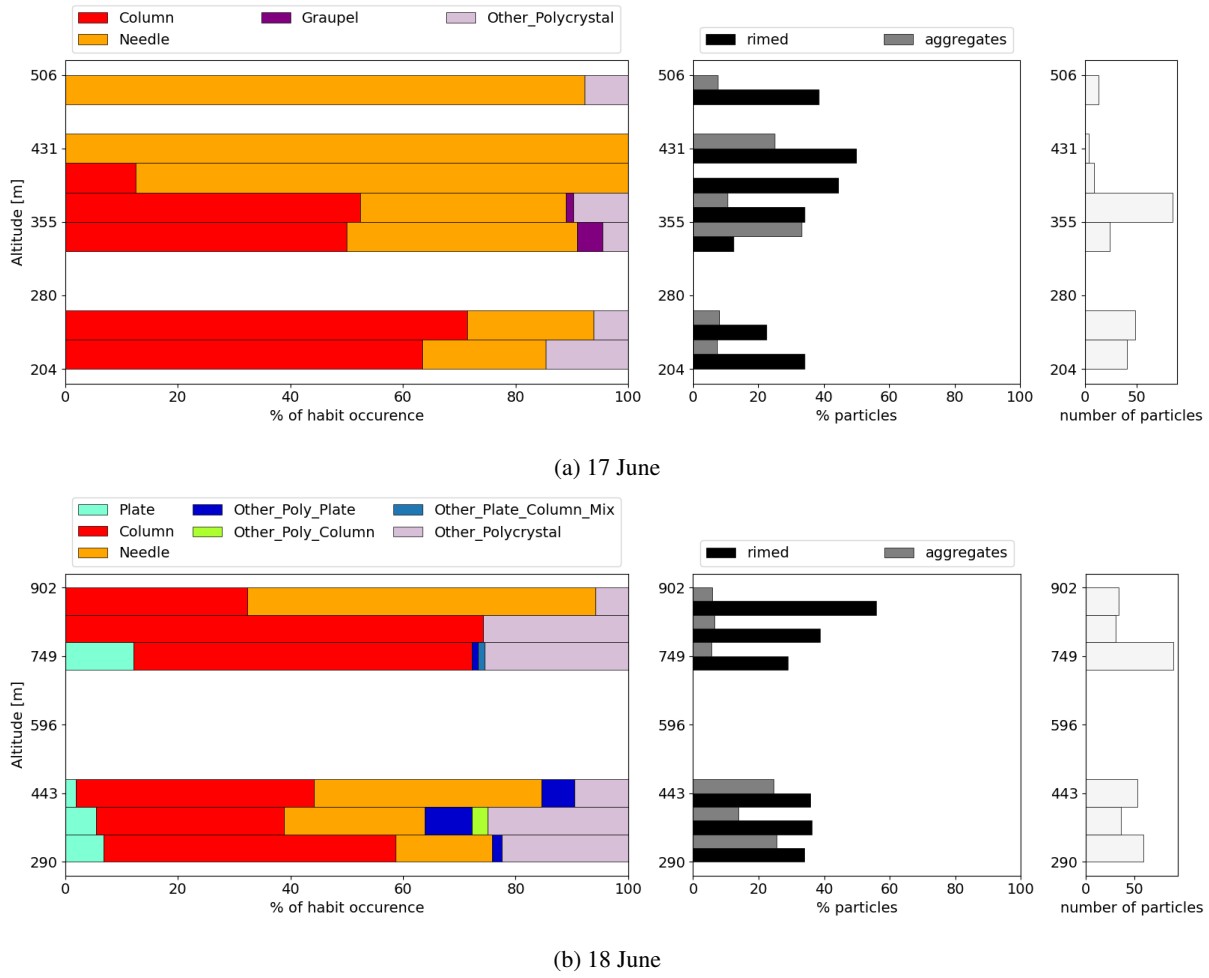

**Figure 11.** Analysis of ice crystal habits during cloud sampling on 17 and 18 June during normal period.

m. The cloud top was observed to vary between 1225 and 1320 m with a very weak temperature inversion around 1°C. The cloud top temperature was -5.3°C.

Figure 12 displays the vertical distribution of mean microphysical properties of both liquid and ice phases. For this case, ice phase microphysical properties were derived for particles with a diameter larger than 200 μm due to observed shattering in the PHIPS and SID-3 probes. In contrast to the other vertical profiles discussed previously, the 18 June case shows that the ice phase dominated both the IWC and extinction at cloud top, even when taking into account the uncertainty in the M-D relations. The ice phase extinction coefficient reached a maximum mean value of 6 km$^{-1}$, which is the highest value observed among the presented cases. Similarly, the IWC had a maximum mean value of 0.08 g m$^{-3}$ when using the BF95 M-D relation, and values seven times lower when using habit-dependent M-D relations assuming all crystals are needles. The mean ice number concentration of crystals larger than 200 μm was up to 5 L$^{-1}$.

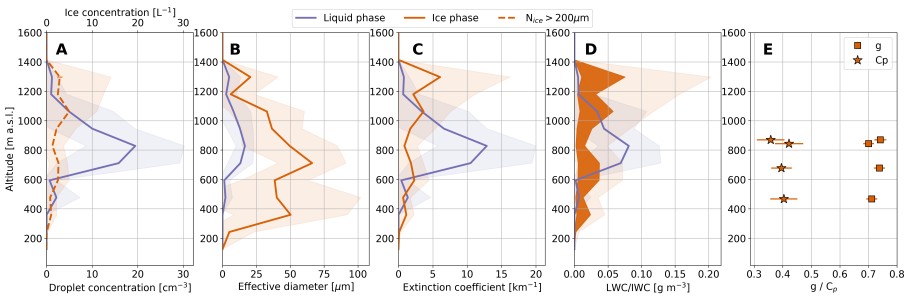

**Figure 12.** Vertical profile of average droplet number concentration (A), effective diameter (B), extinction coefficient (C) and condensed water content (LWC/IWC) (D) performed on 18 June over open ocean during the normal period. The shaded area illustrates the standard deviations. The horizontal error bars in panle A illustrate the statistical uncertainty in $N_{ice>200\mu m}$ calculated using $\sigma(N_{ice>200\mu m})/\sqrt{n}$, where $n$ is the number of counts included in calculation of the mean concentration. In panel D the dark shaded area illustrates the range in IWC when using M-D relations of BF95 (upper limit) and a habit dependent M-D relation of Lawson and Baker (2006) for needles (lower limit). The 10-s average measurement data was binned in altitude bins with bin width of 117 m for calculating the statistics. Cloud sampling was performed between 78.2.1°N and 78.3°N and 5.2°E and 9.9°E.

The liquid phase extinction coefficient and LWC peaked at lower altitudes around 800 m, where maximum mean values of 14 km$^{-1}$ and 0.09 g m$^{-3}$, respectively, were observed. The liquid phase droplet concentrations, LWC, and extinction coefficient were the lowest for all cases with a cloud top temperature around -5°C. This reduction in LWC can probably be explained by the ongoing glaciation of the cloud through the Wegener-Bergeron-Findeisen (WBF) process.

Inspection of the CIP images showed that the increase in ice crystal concentrations near cloud top was caused by predominantly by appearance of columnar type crystal habits, which dominated the upper parts of the cloud. A large fraction, over 20% of the observed columns were rimed according to the PHIPS images (Fig. 11b). In the lower parts of the cloud both CIP and PHIPS showed also compact polycrystal habits. According to habit statistics based on PHIPS images more polycrystals were observed on the 18 June case compared to other days, between 5.9 and 30.6% (Fig. 11b). Comparable to the warm period profiles, also unrimed small faceted columns and plates were observed.

## 4  Asymmetry parameter of mixed-phase cloud ice particles

In order to calculate radiative transfer in clouds, at least three single-scattering properties of cloud particles are required: extinction coefficient, single scattering albedo, and the asymmetry parameter. Of these, the asymmetry parameter is the most sensitive to the assumed particle microphysical properties in the short-wave, particularly for aspherical ice particles whose asymmetry parameter is not well constrained. Here, we derived the ice crystal asymmetry parameter from partial phase functions measured by PHIPS. Since PHIPS detects individual cloud particles, we were able to determine the cloud asymmetry parameter for ice particles only, without interference from supercooled liquid droplets.

Figures 7E, 10E and 12E show the vertical profiles of the ice crystal asymmetry parameters for the cloud cases observed during the warm and normal periods. During the cold period not enough ice crystals were sampled by PHIPS in order to investigate the vertical variability of the asymmetry parameter. The observed ice crystal asymmetry parameters ranged from 0.69 to 0.76, with a mean value of 0.72. In the lower parts of the cloud no vertical trend in the asymmetry parameter was observed, but during the warm period an increase in the asymmetry parameter was seen between $Z_n = 0.6$ and $Z_n = 0.8$. This increase in the asymmetry parameter is linked with a simultaneous decrease in the complexity parameter (Fig. 7E). The fraction of less complex particles can be enhanced in the upper parts of the cloud due to generation of faceted crystals by SIP and by higher fall speeds of larger more complex crystals, which would enhance their concentration in the lower parts of the cloud. As the rimed fraction increased with increasing altitude the decrease in crystal complexity cannot be explained by decrease in riming.

The measured ice particle asymmetry parameters found during ACLOUD are significantly lower than those of idealised faceted hexagonal crystals but also lower than previously found in Arctic mixed-phase clouds. Jourdan et al. (2010) performed measurements with the Polar Nephelometer (PN) in an Arctic nimbostratus cloud during the ASTAR campaign in a temperature range between -1 and -12°C. The authors applied principal component analysis to the volume angular scattering measurements to derive asymmetry parameters for ice particles. The lowest asymmetry parameter found using this method was 0.755 that corresponded to a group of columnar crystals mixed with a small fraction of water droplets - similar habits found in our case. Additionally, Mioche et al. (2017) showed measurements of asymmetry parameters associated with Arctic mixed-phase clouds, where asymmetry parameters below 0.8 were observed for the ice phase.

## 5  Case study of radiative transfer in a single-layer cloud system

Our observations have shown that ice is common in spring and summer-time low level Arctic clouds with cloud top temperatures above -10°C. At the same time, a low ice crystal asymmetry parameter below 0.75 was observed in most parts of the clouds, which is lower than currently assumed by ice crystal optical parameterisations. To investigate the sensitivity of cloud albedo and transmissivity to the choice of ice crystal asymmetry parameter in observed Arctic low-level clouds, we performed radiative transfer simulations using the one-dimensional plane-parallel discrete ordinate radiative transfer solver (DISORT). To evaluate the sensitivity of albedo and transmissivity to ice particle optics, we defined the single-wavelength (532 nm) albedo/transmissivity difference as:

$$\Delta R = R_{true} - R_{mod}, \tag{9}$$

$$\Delta T = T_{true} - T_{mod}, \tag{10}$$

where $R_{mod}$ and $T_{mod}$ are the albedo and transmissivity associated with the cases where ice particles will have modified asymmetry parameter. On the contrary, $R_{true}$ and $T_{true}$ are associated with the case where ice particles are given their measured

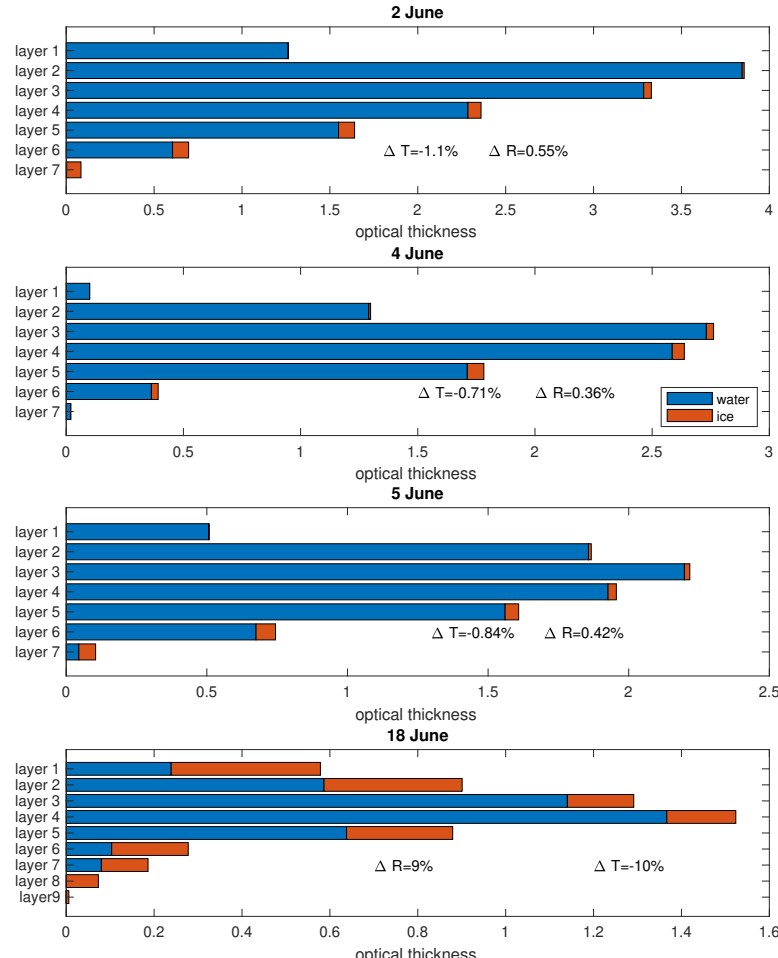

**Figure 13.** The optical thickness for liquid and ice phase for different altitudes within the cloud. The difference in albedo ($\Delta R$) and transmissivity ($\Delta T$) are calculated for modified simulation where ice phase is treated optically like the liquid phase.

asymmetry parameter. The simulation setup is as follows, the surface albedo for the warm period cases are set to be 0.5 to represent the sea ice surface (Stapf et al., 2020), while the surface albedo for the last case is set to be 0.1 to represent the open sea surface (Payne, 1972). The solar zenith angle is set to be 65.9°. For the case studies we have chosen four cloud profiles: the first three cases are cloud profiles over pack ice during the warm period, while the last case is the convection influenced stratiform case over open sea surface. Figure 13 displays the vertical distribution of optical thickness for ice and water within these clouds.

In our first set of simulations, we tested the impact of treating ice particles as if they were liquid droplets ($g\sim0.87$). This approach would be appropriate if cloud optical thickness is known but not the phase composition, and all cloud particles are assumed to be spheres. We found that such an assumption would generally lead to differences in albedo and transmissivity of less than 2% for the warm period cases, where the liquid phase dominated the optical thickness throughout the cloud. Therefore, we conclude that the optical properties of the ice phase had an insignificant effect on cloud albedo and transmissivity during 400 the warm period cases, as long as the cloud optical thickness was preserved.

However, on June 18th, the ice phase dominated the optical thickness in the top layer of the cloud. Therefore, treating the ice phase as liquid resulted in significant differences in albedo and transmissivity, up to ten percent. To investigate this further, we repeated the simulations for June 18th but assigned the ice phase an asymmetry parameter of that of a smooth hexagonal column, which is typically assumed in climate models ($g\sim0.78$) (Fu, 2007). These simulations resulted in albedo 405 and transmissivity differences of only 1.4%. Therefore, we can conclude that for all the case studies shown here, the assumed asymmetry parameter for ice crystals resulted in insignificant changes in albedo and transmissivity as long as the ice crystals were considered to be aspherical.

The simulations presented in this study aimed to isolate the effect of the ice phase optical properties on radiative transfer by keeping the cloud optical thickness fixed. However, in reality, removing the ice phase would likely increase the total cloud 410 optical thickness. This is because the condensed phase would redistribute from fewer and larger ice crystals onto more and smaller droplets, increasing the particle surface area. To fully understand such secondary effects of the ice phase on radiative transfer, more sophisticated simulations would be needed.

## 6 Discussion and conclusions

Quantification of the vertical structure of ice and liquid microphysical properties in Arctic low-level clouds is important to get 415 insight into the microphysical processes governing the cloud lifetime and optical thickness. Information on the ice and liquid optical properties are needed to calculate the cloud radiative effects. Here, we focus our discussion to the cloud ice phase and discuss the potential ice formation mechanisms.

Our vertical profiles showed that relatively high ice crystal number concentrations and IWC (average values up to $56\,\mathrm{L}^{-1}$ and $0.08\,\mathrm{g\,m}^{-3}$) were observed during the warm and normal periods with the coldest cloud top temperature at -6.7°C. Previous 420 observations in the Ny Ålesund region have shown that springtime INP concentrations at -7°C are on average below $10^{-3}$ $\mathrm{L}^{-1}$ with 95-percentile being below $10^{-2}\,\mathrm{L}^{-1}$ (Li et al., 2022). This is in consensus with the general knowledge of INP concentrations in this temperature range (Kanji et al., 2017). Therefore, it is likely that the observed ice crystal concentration in the discussed warm and normal period cases were results of ice multiplication. A likely SIP mechanism in this temperature range is rime-splintering, which is supported by the observations of large rimed needles co-existing with droplets with $D >20$ 425 µm. Also other SIP mechanisms are possible and based on the in-situ data alone it is not possible to confirm or disclose any SIP mechanism.

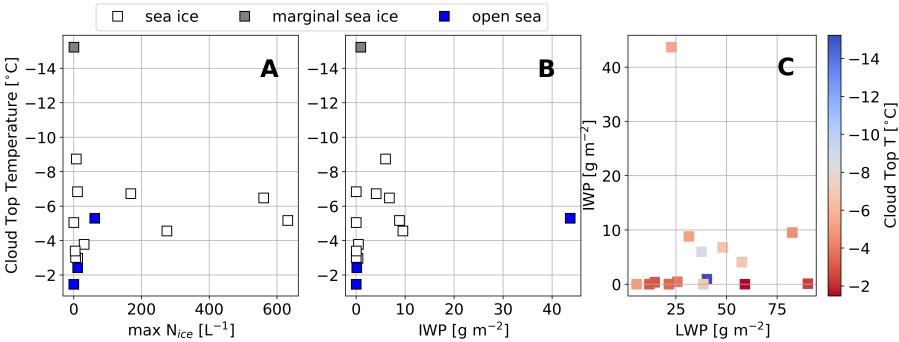

**Figure 14.** Maximum 10-second ice crystal number concentration ($N_{ice}$; panel A) and ice water path (IWP; panel B) as a function of cloud top temperature during ACLOUD for vertical cloud profiles over pack ice (white symbols), marginal sea ice zone (grey symbols) or open sea surface (blue symbols). The correlation between liquid water path (LWP) and IWP is shown in panel C where the symbols are color-coded with the cloud top temperature. The list of vertical cloud profiles can be found in the supplementary material.

It should also be noted that elevated ice crystal number concentrations were not observed throughout the entire vertical extent of the cloud on all days. For example, on June 17th, enhanced ice crystal number concentrations were only observed below 500 m (see Fig. 10). Additionally, on this day, the measured cloud was inhomogeneous with a lower LWC in the upper cloud layer and a higher LWC in the lower cloud layer. This observation could indicate that a higher LWC or larger droplet sizes (see Fig. S29) promoted ice multiplication, as would be the case in rime-splintering or droplet shattering.

To further investigate the occurrence of SIP during ACLOUD, we analysed additional vertical profiles for their maximum 10-second ice number concentration and IWP. Altogether 15 vertical profiles were included in this analysis, from which one was performed over marginal sea ice zone, 11 over pack ice and 3 over open sea surface. A complete list of these cloud profiles can be found in the supplementary material. Figure 14 displays the maximum ice crystal number concentration and IWP of each vertical profile as a function of cloud top temperature. It should be noted that the integrated water paths presented here present the average values for each vertical profile and do not take into account variability in cloud microphysical properties or cloud base and top, which was occasionally observed (see Table 1).

For both the maximum ice crystal number concentration and IWP a clear vertical trend can be seen where up to several orders of magnitude higher number concentrations and IWPs are seen when cloud top temperatures are between -8.7 and -3.8°C. This increase further supports active rime-splintering in the Hallett-Mossop (HM) temperature range and subsequent growth of ice splinters in water saturated environment.

Out of the five warm period cases and nine normal period cases, four and eight, respectively, showed maximum ice crystal number concentrations above 1 L$^{-1}$. This suggests that SIP was frequently occurring in the low-level clouds during both warm and normal periods. In one warm period case, no ice was observed due to a warm cloud top temperature (-1.5°C) outside the HM temperature range. In the one case during the neutral period, the maximum ice crystal number concentration was 0.01

L$^{-1}$, and the cloud top temperature (-5.1°C) was within the HM temperature range. However, the LWP was only 5.5 g m$^{-2}$, and no drizzle-sized droplets were observed.

Half of the profiles with maximum ice crystal number concentrations above 1 L$^{-1}$ also showed IWP above 1 g m$^{-2}$. In
these cases, ice crystals with D>1 mm were observed in the lower half of the cloud. For the profiles with IWP below 1 g m$^{-2}$, ice crystals with D<1 mm were observed. Nevertheless, ice crystals were observed throughout the cloud, except for the cloud top, indicating early stages of glaciation initiated by SIP. SIP in clouds with low IWP and only D<1 mm ice crystals might go undetected by radar remote sensing. This highlights the need for in-situ measurements to investigate the conditions favoring SIP.

Figure 14 also shows that for most of the cases (except 18 June) IWPs above 1 g m$^{-2}$ are only observed when LWP are above 30 g m$^{-2}$. This can indicate that a threshold LWP is needed in stratiform clouds before SIP is efficient in glacification of the cloud. Increasing number of evidence have shown that SIP rate is enhanced in environments with higher LWP, such as in updraft regions (e.g. Lasher-Trapp et al., 2021; Mages et al., 2022).

SIP in Arctic clouds with cloud top temperatures above -8°C have been previously reported by Rangno and Hobbs (2001)
and Pasquier et al. (2022). Rangno and Hobbs (2001) observed ice crystal concentration up to 40 L$^{-1}$ in Arctic stratocumulus clouds in late spring and early summer. Their type III slightly supercooled startiform clouds, containing droplets with D >28 μm and high ice crystal concentrations, explain well our observations during the warm period. Pasquier et al. (2022) explained that high (>50 L$^{-1}$) ice concentrations at temperatures above -5°C were related to SIP by droplet shattering mechanism. In our case superadiabatic LWC profiles could have indicated larger droplets and in all vertical profiles during warm and normal
period drizzle droplets were observed (see Figs. S26-S30). Large droplets could promote either droplet shattering or enhance rime-splintering rate (Mossop, 1985). During M-PACE campaign SIP was not discussed by McFarquhar et al. (2007) but a later modelling study using the Community Atmosphere Model version 6 (CAM6) by Zhao et al. (2021) showed that inclusion of SIP in the model was important for improving the model representation of the observed cloud situations. However, other studies do not see enhancement in IWP in clouds with cloud top temperature above -10°C. For example, in Mioche et al. (2017)
no increase in IWP in the HM temperature range was found for late spring Arctic clouds over open ocean.

Even though SIP is not a new phenomenon in the Arctic, our results show that SIP can be observed also in low-level clouds over pack ice in the late spring and early summer. Ice phase has an important role in governing the cloud lifetime and cloud albedo through WBF process and scavenging on droplets in riming process. Heterogeneous ice nucleation parameterisations in climate and general circulation models predict an decrease in ice crystal number towards higher temperatures (e.g. Meyers
et al., 1992; Phillips et al., 2013). However, increasing observational evidence (present study included) is showing that in the Arctic a negative correlation with ice crystal number concentration and temperature does not apply for all of the warmer clouds (see e.g. Gultepe et al., 2001). It is important that general circulation and climate models correctly capture the ice phase of low-level mixed-phase clouds, for which inclusion of SIP in the models is needed. However, since ice multiplication mechanisms are still poorly understood (Field et al., 2006), detailed in-situ observations of the ice phase are important for
testing and constraining SIP parameterisations in the Arctic environment. In these lines, a few studies have already studied SIP in Arctic clouds observed during field campaigns (e.g. Fridlind et al., 2007; Zhao et al., 2021; Sotiropoulou et al., 2020). All

these studies highlighting the importance of SIP, yet open questions remain, such as how many SIP mechanisms are needed to predict the observations and what is the contribution of the different SIP mechanisms.

Ice phase changes the cloud optical properties due to the fact that aspherical ice crystals have different single-scattering properties compared to spherical droplets. For example, in the visible wavelengths an aspherical ice crystal has a lower asymmetry parameter compared to a droplet but the magnitude of the asymmetry parameter reduction is sensitive to the exact morphology of the ice crystal. We asked in this study, how important is the exact magnitude of the ice phase short-wave asymmetry parameter for the cloud albedo and transmissivity in the presented cloud cases. Simulations showed that in the typical cases, where the cloud top in almost completely composed of liquid phase and where ice phase is not a major contributor to the cloud extinction, the exact magnitude of the ice asymmetry parameter is not important for the radiative transfer. However, if the ice phase dominated the cloud top extinction, as was the case on 18 June, a 10% underestimation in the albedo and transmissivity was seen, when assuming spherical particles. Yet, this underestimation was reduced to only 1.4% when the modified simulations used the asymmetry parameter of pristine hexagonal columns. These results highlight that ice phase in the observed clouds did not have a significant contribution to the cloud transmissivity and albedo. Only when ice phase is found at the top of mixed phase clouds can the treatment of ice optical properties become important. However, it should be kept in mind that the presented simulations only considered the direct optical effects of the ice phase and not secondary effects such as modification of the cloud liquid phase properties in the absence of the ice phase.

## 7  Summary

As clouds in the Arctic have the potential to insert a positive feedback in a warming climate, it is important to increase the knowledge of the vertical distribution of the cloud water content and phase. Measurements of late spring and summer time stratiform clouds over pack ice, marginal sea ice zone and open water performed during the ACLOUD campaign showed that relatively high ice particle number concentrations up to $57 \, \text{L}^{-1}$ are observed in cases where cloud top temperatures are between -3.8 and -8.7°C. This elevation in ice crystal number can likely be linked with secondary ice production. Still, the condensed water path is dominated by the liquid phase especially at the cloud top in most of the studied cases except in one case study of a system with embedded convection where ice extinction exceeded the liquid extinction. Simultaneous measurements of ice optical properties showed that a relative low asymmetry parameters between 0.69 and 0.76 can be associated with the mixed-phase cloud ice crystals. However, it was shown with radiative transfer simulations that the exact choice of ice crystal asymmetry parameter did not significantly impact simulated cloud albedo and transmissivity in the studied cases.

The presented results highlight that there exists a complicated and not negatively correlated relationship between cloud top temperature and ice crystal concentration. In order to accurately predict Arctic warming, it is important that models capture the cloud microphysical processes in Arctic clouds. In-situ observations provide an important basis for testing and improving microphysical parameterisations.

*Data availability.* The observational data from the ACLOUD campaign are archived on the PANGAEA repository and can be accessed from the DOIs given in the references.

**Appendix A: Machine learning algorithm to detect co-incidence artefacts in SID-3 scattering patterns**

The SID-3 camera has a theoretical coincidence sampling probability of 1% for a particle number concentration of 103 $\mathrm{cm}^{-3}$ (Vochezer et al., 2016). Coincidence sampling typically occurs for two droplets so that instead of a scattering pattern with concentric rings (see Fig. A1a) a distorted scattering pattern, e.g., a "kidney" shaped shadow, is recorded (see Figs. A1c or A1d). Coincidence patterns are always classified as aspherical particles by the automatic classification scheme, so that in a

520 mixed-phase cloud environment a coincidence sampling probability of 1% would lead to a significant overestimation of ice concentration. However, due to the high-resolution of the SID-3 2-D scattering patterns the coincidence affected patterns are easily identifiable by human eye. Since manual re-classification of coincidence patterns is very time-demanding as SID-3 typically acquires in the order of ∼100.000 images per flight, here, we have trained a deep learning neural net that classifies particles measured by SID-3 based on their 2-D scattering pattern.

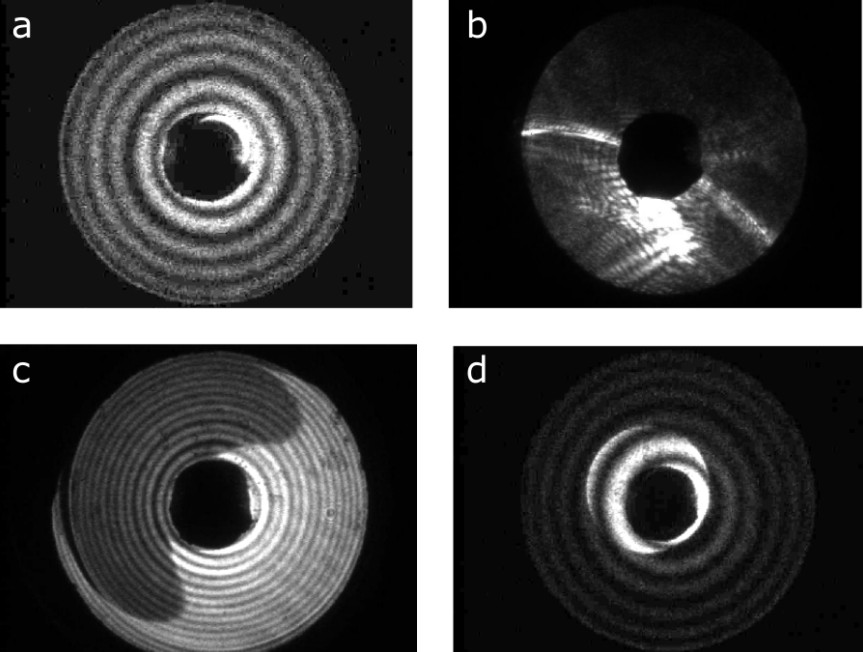

**Figure A1.** Measured SID-3 scattering patterns of exemplary particles: droplet (a), ice particle (b), coinciding droplets (c,d).

## A1 Data Basis

The data used to train and validate the net consists of a subset of i) previously manually classified particles from the ACLOUD campaign and ii) particles from cloud segments where SID-3 flew in pure ice or pure liquid cloud clouds during the CIRRUS-HL and ACLOUD campaigns.

Set i), the manually classified data, consists of 460.000 droplets, 75.000 coincidence events and 2488 ice particles. Only images with mean-intensity between 10 and 50 were manually classified (about 50% of all ACLOUD data) to remove coincidence artefacts from crystal complexity analysis performed in Järvinen et al. (2018). As this set is heavily biased towards liquid, the following subset of 4.000 particles was selected at random out of each category: 1.000 particles that were both classified as liquid by the discrimination algorithm and the manual classification (true liquid) and 1.000 particles that were classified as ice by the algorithm but identified as coincidence events by manual classification (false ice) and 2.000 ice particles that were identified by both the algorithm and manual classification (true ice).

Set ii) consists of cases where SID-3 flew in a pure ice cloud (CIRRUS-HL flight RF12) and a pure liquid cloud (ACLOUD flight on 17.6.2017). To match the numbers of subset i), 2.000 particles were selected at random from each flight. During the ACLOUD liquid case, an estimated 14% of all images were coincidence droplets. Note that these data are not restricted based on the image mean-intensity range as the manually classified particles in i).

Combined, subsets i) and ii) consist of a total of 8.000 particles, equal parts liquid and ice, which are split in half as training and validation data sets. Both data sets consist of in total 4.000 particles each, 1.000 manually classified ice particles, 1.000 ice particles recorded in a cirrus cloud, manually classified 500 true droplets and 500 coincidence droplets, 1.000 droplets recorded in a warm (T>0°C) cloud. These particles are chosen at random out of the above mentioned data set. The training and validation data set are completely disjunctive, i.e. the net is validated on data it has not yet seen before during training.

The single particle 2-D scattering pattern is saved as a 780x582 px 8-bit image as a .jpg file. Due to limited GPU RAM and to reduce computation time of the neural net, each image was scaled down by x1/10 from 780x582 px to 78x89 px.

## A2 Neural Net

Based on this data, a deep learning neural net was trained. It was set up using the embedded deep learning environment in MATLAB (MATrix LABoratory) consisting of a 2-D [59x78] image input layer, a 2-D convolutional layer with 64 filters of size [5 5], a batch normalization layer, a rectified linear unit (ReLU) layer, a 2-D max pooling layer with pool size [2 2] and stride [2 2], two fully connected layers with output sizes 10 and 2, a softmax layer and finally the binary classification layer (1/0, corresponding to liquid/ice). The solver is a stochastic gradient descent with momentum (SGDM) optimizer with initial learning rate of 0.05 and a maximum number of epochs of 15.

## A3 Discrimination Accuracy

Fig. A2 shows the confusion matrix for the validation data set. The net classifies 98.8% of all droplets correctly (i.e., in a liquid cloud 1.1% of particles are classified as ice). For ice particles, the accuracy is 99.1%. This a better accuracy compared to the

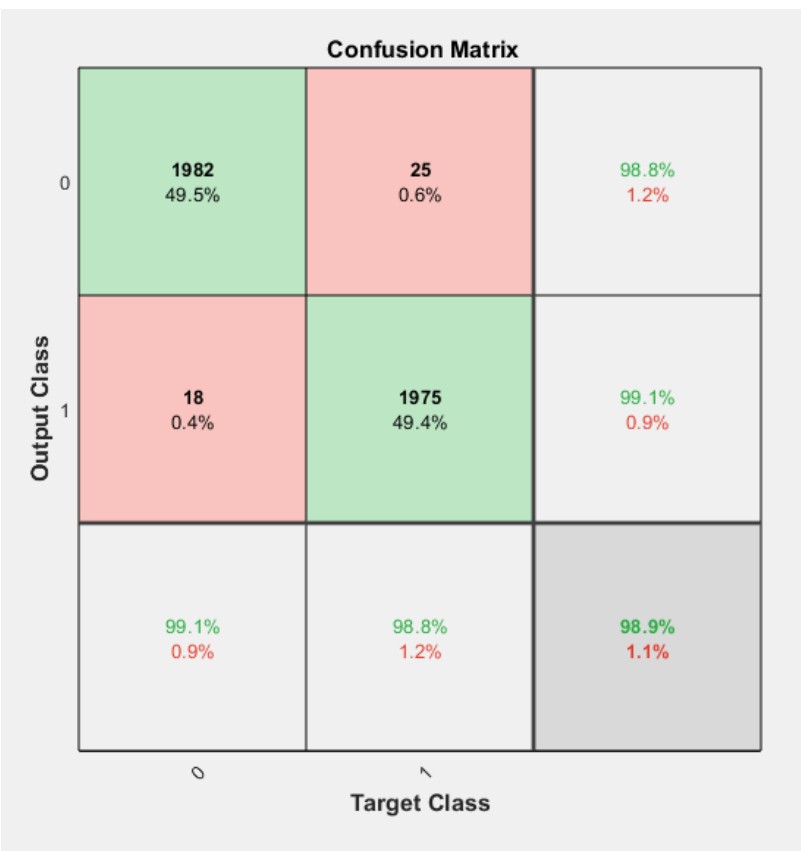

**Figure A2.** Confusion matrix that visualizes the classification accuracy of the phase discrimination neural net based on the validation data set.

original algorithm which had a misclassification rate of 2%-20%, depending on particle density and hence coincidence rate. Interestingly, the true coincidence rate observed in the training data set was significantly higher than the theoretical expected coincidence rate of 1% for the expected droplet concentrations. This indicates that droplets might be located in concentrated

pockets within the cloud increasing the coincidence probability, or, that shattering of larger ice particles might cause shattering fragments to coincide in the camera field-of-view.

A misclassification rate of 1.1% can still result in a significant overestimation of ice by a factor of 2 to 10. Therefore, we combined the neural net analysis with manual inspection in the following way: if a particle was classified as spherical (liquid) by either the original algorithm or the neural net, it was considered liquid. During ACLOUD the remaining number

of 2-D scattering patterns where both or either the original algorithm or the neural net classified a particle as aspherical (ice) was typically low enough (around 100 - 1.000 per flight) so that those 2-D scattering patterns were manually inspected and reclassified if necessary. The SID-3 ice concentration shown in this paper can, therefore, be considered as the lower limit for ice concentrations below 50 μm.

*Author contributions.* MS and EJ operated the SID-3 and PHIPS instruments during the ACLOUD campaign. OJ, RD and GM operated the
570 CIP instrument during ACLOUD. EJ, FN, FW and MS analysed the SID-3 and PHIPS data and OJ, RD and GM analysed the CIP data. GZ
performed the radiative transfer calculations. All were contributing to the interpretation of the results. EJ wrote the manuscript with help of
all co-authors.

*Competing interests.* Martin Schnaiter and Emma Järvinen are members of schnaiTEC GmbH that manufactures a commercial version of
the PHIPS instrument. Martin Schnaiter is part-time employed by schnaiTEC GmbH.

*Acknowledgements.* This work received funding through the Helmholtz Association's Initiative and Networking Fund (grant agreement
no. VH-NG-1531), the Helmholtz Association research program "Atmosphere and Climate". MS acknowledges funding from the German
Research Foundation (grants SCHN 1140/3-1 and SCHN 1140/3-2). FW acknowledges funding by the German Research Foundation (DFG
grant JA 2818/1-1). The LaMP acknowledges the support of the (MPC2-EAn n°1224) project funded by the Institut Polaire Paul Emile Victor
(IPEV) and the Pollution in the Arctic System (PARCS) project funded by the Chantier Arctique of the Centre National de la Recherche
Scientifique-Institut National des Sciences de l'Univers (CNRS-INSU). The authors are grateful to AWI for providing and operating Polar
6 during the ACLOUD campaign. We thank the crew of the Polar 6 and their technicians for the excellent technical and logistical support
and the scientists who operated the instruments during the science flights. We gratefully acknowledge the two anonymous reviewer's, who's
comments improved this manuscript significantly.

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
