# Peer review of "Investigating the vertical extent and short-wave radiative effects of ice phase in Arctic summer-time low-level clouds"

_Atmospheric Chemistry and Physics, 2022_

## Referee Comment (RC2)

Summary

This work using ACLOUD aircraft observations studies BL Arctic cloud microphysics profiles based on various in-situ probes. Overall, this kind of studies is needed to improve NWP model simulations and predictions. Authors looked into BL mixed phase clouds over various surfaces and found out that Ni is about less than 35 L-1 and IWC was between 0.003 and 0.08 g m-3. Cloud top T was measured between -3.8 and -8.7C and likely resulted in Ice Multip process but this finding was speculation and no indicators or metrics are given. Not clear, why ice phase had minor impact on transmissivity? Cloud phase as stated is critical for RTM and simulations but this study did not show any RID or even RHw where the supercooled droplets or mixed phase clouds exist.

In addition, intro is found to be very weak, and only certain people work indicated rather than broader knowledge  supposed to be given and referenced.

No proper discussion section is provided, and conclusions are mostly speculative.

Based on the above points and issues provided below, I suggest major corrections for this study.

Major/minor issues:

Line 22; specifically please see Gultepe et al BL clouds/ice fog work for Arctic clouds, this work is directly related to your work here.

Ln35-40; see Ni-T for ice clouds based on various studies, and Gultepe et al showed that no trend in Ni-T (Intern J. of Climate)

Ln50; see Gultepe et al  for SIP and ice crystal splintering issues (AMS met Monographs2018; Arctic ice cloud studies)

Ln58; provide a ref for Ice fog/BL clouds high Ni values found in Arctic clouds, Lawson et al and Gultepe et al (Atmos Res review, 2017)

Ln68; why suddenly g became important here?

Ln116; why cip not used for 15 micron bins? See Gultepe et al grey probe used for BL ice clouds and drizzle.

Ln152; why assumed as spheres? See Gultepe et al Ice fog review in Atmos Res 2017

Eq. 4; what is Qeff? And what wavelength considered here? (See AMS Bull Arctic  Ice Fog campaign)

Fig. 2; where is RHw? If not saturated why we see SIP particles? Why not show RID?

Same fig; you cant just plot wind speed like this, see Gultepe et al 1990 and 1995 Arctic cirrus studies.

Fig. 3; how did you filetered data to get solid lines???? From what? Aircraft?

Fig 5; how did you removed the snow from the analysis? Snow versus cloud?

Fig 6; same issue?

Ln280; Gultepe et al (Atmos Res review, Arctic ice particles were not irregular), they showed it clearly also.

Fig 09; how did you interprete data? These are just selected ones, explain it.

Fig 12; why you normalize the data??? Show T also, o idea what is T?

Fig 13; a has some relationship but not others, why?

Section 6; not clear and results are not meaningful to me, please discuss and compare with others.

Parag 425/conclusions; very well known statements, and I see that conclusions need to be improved and referenced for the knowledge given in the paper.

Finally, how did you discriminate SIP from the prime ice crystals? What is the basis?

I suggest major revisions for this work, and see it again.

---

## Author Comment (AC1)

**Author Responses to Anonymous Reviewer #1**

This paper presents some interesting results on the vertical profiles of mixed-phase clouds measured during the ACLOUD campaign, finding results similar to those found from previous studies of this cloud genre in that liquid dominates the upper parts of clouds with ice phase observed in greater amounts in the lower parts of the cloud. This study is worth publication because it presents data from a new location and presents additional data about the measured asymmetry parameters that have not typically been measured during prior campaigns. They also present radiative transfer simulations to claim that the ice phase only has a minor effect on cloud transmissivity and albedo. The paper is well written, technically sound, and easy to follow. Thus, I am recommending that the paper be published, but there are some points that need to be clarified before the paper is published.

*We thank the Reviewer for this positive general comment. Below we give detailed responses to the Reviewer comments and explain the modifications we made to produce the revised version of this manuscript.*

I have some uncertainty in how the flight profiles are used to construct the vertical profiles of cloud properties as a function of normalized altitude. It is stated that "vertical profiles were measured in-situ either by flying a double-triangle pattern, where altitude was changed at the outer vertices, or by flying stacked horizontal legs….[with sampling lasting] between 7 to 10 minutes." However, it is unclear to me how many different altitudes were sampled during the stacked horizontal legs. Can you list the number of levels that were flown in Table 1?

*The vertical profiles are based on 10-s averaged cloud microphysical data. To create the averages and standard deviations as a function of normalised altitude the 10-s data was binned using the bin limits shown in table 1. This includes five bins that were inside the cloud. Since the number of bins is larger than the number of horizontal legs (typically between 3 and 5), some bins contain better statistics than others.*

*Table 1: Normalised altitude bins.*

| bin | Old lower Zn | Old upper Zn | Revised lower Zn | Revised upper Zn |
|:---:|:---:|:---:|:---:|:---:|
| 1 | -0.3 | -0.1 | -0.6 | -0.3 |
| 2 | -0.1 | 0* | -0.3 | 0 |
| 3 | 0 | 0.25 | 0 | 0.25 |
| 4 | 0.25 | 0.45 | 0.25 | 0.45 |
| 5 | 0.45 | 0.65 | 0.45 | 0.65 |
| 6 | 0.65 | 0.85 | 0.65 | 0.85 |
| 7 | 0.85 | 1 | 0.85 | 1 |
| 8 | 1** | 1.05 | 1** | 1.05 |
| *Bin mean was not used in plotting but Zn = 0 **Bin mean was not used in plotting but Zn = 1 | | | | |

In Figure 3 and 4 can the specific altitudes used in the construction of these plots be indicated?
*We think that adding the altitude limits to Figs. 3 and 4 would have a negative influence to the readability of the figures, which contain rather high information load already. However, we added in Supplementary Information (SI) figures that show how the average profiles were constructed from the 10-s microphysical data.*

*These figures show (i) the flight pattern, (ii) time series of height and LWC/IWC and (iii) scatter plot of 10-s average LWC and IWC as a function of altitude. The panel (c) also shows the altitudes that were used as bin edges.*

*When producing these vertical profile summary figures, we noticed that the binning below the cloud base was not optimal. We adjusted the bins so that the below cloud bins are now from -0.6 to -0.3 and to -0.3 to 0. The changes are illustrated in Figure 3. This change in below cloud bin edges did not have consequences for the conclusions.*

[Figure]

*Figure 1: Illustration of the flight pattern as a function of latitude, longitude and height (first panel), time series of the altitude and LWC (second panel) and 10-s average LWC (blue) and IWC (red) as a function of altitude. The grey lines illustrate the altitude bin edges used to calculate statistics.*

[Figure]

*Figure 2: Same as Figure 1 but for the 2 June case (old height bins). The darker horizontal lines represent the cloud top and base altitudes.*

[Figure]

*Figure 3: Same as Figure 1 but for the 2 June case (new height bins).*

*The discussion of how flight data is used to derive the vertical profiles is improved based on the suggestions of the Reviewer. This discussion takes place in the beginning of the Sec. 3. Figure captions were also modified so that the height bins are stated. Also, as suggested by the Reviewer, we listed the number of horizontal legs in Table 1.*

Many other studies examining cloud profiles as a function of normalized altitude are based on measurements obtained during ramped ascents and descents so that 1) observations are obtained at multiple altitudes within the same cloud; and 2) there is clear knowledge of base and top due to when the cloud breaks out of cloud.
How many altitudes were sampled during a typical flight?
*Typically between 3 and 5 (as now stated in Table 1).*

How much did cloud base and top vary during the flights, because that would seem to be a major uncertainty in the calculation of the normalized altitude? How were the base and top determined? What is their uncertainty and how does that translate into an uncertainty in Zn?
*The cloud base and top were originally determined based on 10-s average LWC data. The determination of cloud top and base was revised by using 1-s average LWC data, which also enabled closer inspection of periods, when cloud was entered and exited (i.e. LWC threshold of $0.01 \text{ g m}^{-3}$ was crossed). The variation in cloud base and cloud top values were added to Table 1. Also cloud top temperatures were revised but the effect was minimal (maximum of 0.1°C).*

*The re-inspection of the cloud base and cloud top values led to the following changes:*

**27 May**
*Cloud base and top were revised using 1-Hz data. As the cloud was sampled with one descent, we could not determine uncertainty of these values.*

**2 June**
*Using 1-Hz observations had a significant effect on cloud top value, which changed from 470 to 440 m. A second cloud layer was seen at 530 m, with low LWC around 0.17 g m-3, but as this cloud was sampled when leaving the study area, we do not consider this as part of the vertical profile. The new value for cloud top changed the observed liquid vertical profile so that LWC maximum is now found at $Z_n>0.8$. The cloud base was observed to vary between 177 and 201 m. An average value of 189 m was used as cloud base and error bars were added to Figs. 6 and 7 to indicate the uncertainty in $Z_n$.*

**4 June**
*The cloud base was observed to vary between 93 and 103 m. The cloud top was exited and re-entered five times during this vertical profile. The observed cloud top height varied between 374 and 433 m. We used the maximum value of 433 m to calculate Zn. Error bars were added to Figs. 6 and 7.*

**5 June**
*The cloud base was entered only once so uncertainty in the cloud base height could not be determined. The cloud top was exited and re-entered three times during this vertical profile. The observed cloud top height varied between 425 and 445 m and the average of these two was used for calculating Zn. Error bars were added to Figs. 6 and 7.*

**Normal period: 17 and 18 June**
*The normal period had more inhomogeneous cloud fields compared to the warm period. This is reflected in larger variation in the observed cloud top height and base values, as seen in Table 1. For example, the large variation on 17 June was caused by a thin stratus layer that was measured above the main cloud having the top at 934 m. Due to the inhomogeneity of the cloud fields, we agree, that representing the normal period cases as a function of $Z_n$ would result into large uncertainties. We decided to show the normal period cases as a function of (non-normalised) altitude in the revised version. One figure with both liquid and ice phase microphysical properties*

*is now shown for each vertical profile, where the average liquid and ice microphysical properties were calculated for equidistance horizontal bins. For 17 June a bin height of 52 m was used, and for 18 June over open ocean 117 m. The figures and the discussion of the figures was revised as well. We also decided to only focus on those vertical profiles that included ice phase in the revised version so we excluded the 18 June case over pack ice from this section.*

A measure of uncertainty, the standard deviation, is provided in the plots of microphysical quantities versus normalized diameter. How does the uncertainty in the calculated microphysical quantities compare against this standard deviation? A good measure of uncertainty due to statistical counting is provided as proportional to the square root of the number of counts of particles that were counted in each of the size bins. Given that PHIPS has poorer sampling statistics than other microphysical probes, are there adequate statistics in 10 s to calculate N(D)? This should be quantified.

*We investigated the statistical counting uncertainty in liquid and ice phase N(D) values calculated for the vertical bins shown in Figs. 4, 6,7,10 and 12. The statistical uncertainty was calculated for each cloud probe separately for each height bin. For the PHIPS probe the statistical counting uncertainty in ice concentrations varied between 4 and 58%. For the CIP the statistical uncertainty in N(D) was typically a few percent but during the cold period a maximum statistical uncertainty above 100% was calculated for about half of the height bins.*

*The SID-3 phase separated size distributions are calculated by multiplying the total number concentration by the ice and droplet fractions that are calculated by analysing 2D scattering patterns acquired for a sub-sample of particles. In this case the uncertainty in ice and droplet concentrations was estimated using Clopper– Pearson confidence limits (Vochezer et al., 2016) that take into your account the number of successes (e.g., number of detected ice patterns), and the total number of 2D scattering pattern images under consideration. Using this method to estimate the SID-3 statistical uncertainties gives values below 1%.*

*Total statistical uncertainties for ice number concentration were calculated using error propagation and added as horizontal error bars in all figures showing $N_{ice}$. The same was done for droplet number concentrations. It can be seen that the statistical uncertainty is well below the observed standard deviation. In most cases, the statistical uncertainty is so small that the horizontal error bars are not visible. The statistical uncertainty in ice number concentrations >50% is observed when CIP concentrations are below 0.03 $L^{-1}$ or when the PHIPS concentrations are below 2 $L^{-1}$.*

*Following this investigation, we also included the SID-3 uncertainty estimation to Sec. 2.2: "From the numbers of observed spherical and aspherical 2-D scattering patterns the fractions of spherical and aspherical particles are derived. Multiplication of those number-based fractions with the total particle size distribution yields phase-specific particle size distribution. The uncertainty due to the fact that the imaged particles are a subset of all sampled particles can be estimated from the Clopper–Pearson confidence limits as discussed in Vochezer et al., 2016."*

Similarly, a better description of the uncertainty in the calculated LWC and IWC should be provided. Since the calculated IWC from the PHIPS and CIP is based on Brown and Francis (1995), there could be very large errors depending on the mixtures of habits that are actually present at a singe time. Were there any sensitivity studies conducted where either the Baker/ Lawson (2004) technique or application of habit-dependent mass-dimensional relationships explored?

*It is true that the choice of mass-dimensional (M-D) relationship does have an effect on the IWC retrieval. We performed a sensitivity study to investigate this effect by re-calculating the IWC using the following mass-dimensional relationships:*
- *McFarquhar et al., 2007 (Arctic MPC; M07)*
- *Revised version of Mitchel et al., 1990 by Lawson and Baker, 2006, (LB06) for*
    - *All habits*
    - *Needles*
    - *Rimed needles*
    - *Plates*

*The habits were chosen based on the typical habits observed by PHIPS: needles and rimed needles for warm and neutral period and plates for cold period.*

*Figure 4 shows the results of the IWC comparison between Brown & Francis, 1995 (BF95) and other M-D relationships. It can be seen that compared to BF95 all other M-D relationships predict lower IWCs. The M07 and LB06 (all habits and hexagonal plates) are lower by 40 to 60 %. The lowest IWCs, by around 85%, are predicted assuming that the ice crystals are composed of purely (rimed) needles. This highlights the general issue in retrieving bulk microphysical properties, like mass, using particle size.*

*We added a section (S2) to SI showing this sensitivity study. We also added a discussion of different M-D relations to Sec. 2.3.2:*
*"IWC was calculated from PHIPS and CIP measurements using the mass-dimensional (M-D) relations. Since there are several M-D relations in the literature depending from the cloud type and mixture of habits we performed sensitivity studies using M-D relations from Brown and Francis (1995), McFarquhar et al. (2007) and habit-dependent M-D relationships measured by Mitchell et al. (1990) and revised by Lawson and Baker (2006). From the habit-dependent M-D relations the following habits were chosen that represent the habits observed by PHIPS: needles, rimed needles, hexagonal plates and mixture of all habits. The highest IWC was retrieved using the M-D relation be Brown and Francis (1995) and 40 to 60% lower IWC was retrieved using M-D relations by McFarquhar et al. (2007) and habit mixtures of all habits and plates by Lawson and Baker (2006) (Fig. S19). The lowest IWCs (by 85% compared to Brown and Francis (1995)) was retrieved for needles and rimed needles".*

*The figure of ice phase vertical profiles were modified so that they show now IWC ranges instead of single IWC retrieval. The IWC range for the warm and neutral periods (dominated by (rimed)*

[Figure]

*Figure 4: Comparison between IWC from BF95 and IWC calculated using M-D relationships by McFarquhar et al., 2007 (blue), Baker & Lawson, 2006 for all habits (red), for needles (yellow), for rimed needles (purple) and for hexagonal plates (green). Linear regression without intercept was performed.*

*needles) were calculated using the BF95 and BL06 needle M-D relations. For the cold period the IWC range was calculated using BF95 and M07 M-D relations.*

*The statistical uncertainty in 10-s LWC measurement is below 10%, whereas a larger source of uncertainty comes from the precision of the measurements. If we assume a typical precision of 10-30% for light scattering instruments in concentration (Baumgardner et al., 2017) we have an uncertainty of the same magnitude for LWC. We added a sentence about this in Sec. 2.3.2: "Since light scattering instruments typically have a systematic measurement uncertainties between 10 and 30% in concentration, we consider the LWC to have a systematic uncertainty up to 30%."*

And, finally, what are the uncertainties in the calculated vertically integrated water paths? The calculation of a water path from measurements that are collected on ramped ascents/descents is more straightforward as the Delta Z to multiple the measured IWC is straightforward. However, if the vertical profile is computed by combining data from several horizontal legs at specific altitudes, the computation is not quite as straightforward. How much variability in the water paths were noted?

*It is true that it is more straightforward to calculate integrated water paths using measurements collected on ramped ascents/descents than using profiles with horizontal stacked legs. It is also not possible to meaningfully estimate the variability in integrated water paths, when flying a double triangle pattern, since the four outer points do not necessarily cover the full extend of the cloud. When discussing the integrated water paths we added a note that the LWP and IWP represent average values for the whole sampling area and that there is likely variability within the sampling area that we do not quantify. This is a reasonable statement as the conclusion we want to make from this analysis are purely related to occurrence of SIP: at which cloud top temperatures the IWPs are significantly above 0 g m$^{-2}$.*

Finally, it is good that the specific definition to compute re is noted. It might be good to reference McFarquhar and Heymsfield (1998) to emphasize that there are several different definitions that can be used.

*We added this reference.*

Also, I think it is more typical to calculate the effective radii separately for the water droplets (rew) and ice crystals (rei). Why was a single effective radius corresponding to both species used?

*The effective radius was calculated separately for ice and liquid phase. To make this more clear in the text we modified Sec. 2.3.3 as following:*

*"For calculating the effective radius ($r_e$) several definitions are available in the literature (McFarquhar and Heymsfield, 1998). Here the following definitions was used to calculate the effective radius for liquid ($r_{e,w}$) and ice phase ($r_{e,i}$)*

$$r_{e,w} = \frac{3LWC}{2\rho_w \beta_{ext,w}}$$
$$r_{e,i} = \frac{3IWC}{2\rho_i \beta_{ext,i}},$$

*where $\rho$ is the bulk density of water or ice. "*

For the habit classification, can you add a figure that shows examples of each of the habits used in the classification scheme? That provides important observational evidence to assess how well the imaged crystals match these idealized crystals.

*We added a figure summarising the habit classification scheme and figures showing exemplary ice crystals in the SI. Section S1 now gives a detailed summary of the PHIPS habit classification scheme.*

In general, there should be greater quantification of the uncertainties.

*Measurement uncertainties (both statistical and systematic) in total concentrations and condensed water contents are now better quantified in the figures and in the presentation of results (please refer to previous reply). Furthermore, we have given attention that measurement uncertainties are*

*also taken into account in the discussion of the results, in comparison with previous studies and when making conclusions.*

The section 6 on the case study of radiative transfer in the single-layer cloud system is a nice addition to the paper. However, its briefness and lack of detailed discussion makes it seem like this section was almost an afterthought to the paper. I think that this section should be given more prominence in the paper.
*We are pleased that the Reviewer acknowledges this section, and agree that the radiative transfer simulations should be better incorporated to the rest of the paper and that the discussion of the results should be elaborated. Therefore, this section was revised.*

*We want to highlight that also interpretation of the simulation results was revised. In the old manuscript version it was written "To investigate the role of ice phase for the radiative transfer in an Arctic summer-time low-level clouds…", but in the revised version this sentence was corrected as "To investigate the sensitivity of cloud albedo and transmissivity to the choice of ice crystal asymmetry parameter in observed Arctic low-level clouds…". This revision is important as in the simulations we keep the cloud optical thickness fixed, which is not the case in real clouds as prohibiting ice formation likely would lead to higher liquid phase extinction. This constraint is made more clear and discussed in the last paragraph of the section 5 (previously 6).*

*Based on this commend and the general comment from Reviewer #2, we made additional structural change in the manuscript by removing some of the discussion from section 3 and the whole section 4 (LWP/IWP) and merging these into a new section 6 called "discussion and conclusions". The discussion section also contains more detailed interpretation of the results from the radiative transfer simulations and links these results to results from section 3.*

*### References*

*Baumgardner, D., Abel, S. J., Axisa, D., Cotton, R., Crosier, J., Field, P., … Um, J. (2017). Cloud Ice Properties: In Situ Measurement Challenges. Meteorological Monographs, 58, 9.1-9.23. https://doi.org/10.1175/amsmonographs-d-16-0011.1*

---

## Author Comment (AC2)

**Author Responses to Anonymous Reviewer #2**

Summary

This work using ACLOUD aircraft observations studies BL Arctic cloud microphysics profiles based on various in-situ probes. Overall, this kind of studies is needed to improve NWP model simulations and predictions. Authors looked into BL mixed phase clouds over various surfaces and found out that Ni is about less than 35 L-1 and IWC was between 0.003 and 0.08 g m-3. Cloud top T was measured between -3.8 and -8.7C and likely resulted in Ice Multip process but this finding was speculation and no indicators or metrics are given. Not clear, why ice phase had minor impact on transmissivity? Cloud phase as stated is critical for RTM and simulations but this study did not show any RID or even RHw where the supercooled droplets or mixed phase clouds exist.
In addition, intro is found to be very weak, and only certain people work indicated rather than broader knowledge supposed to be given and referenced.
No proper discussion section is provided, and conclusions are mostly speculative.
Based on the above points and issues provided below, I suggest major corrections for this study.

*We acknowledge the general comment from the Reviewer. Our study found relative high ice crystal concentration (>10 L-1) in relatively warm single layer stratocumulus clouds with cloud top temperatures above -9°C. Even without simultaneous measurements of ice nucleating particles (INPs), with the current knowledge of INP concentrations it is safe to assume that these ice crystals likely resulted in ice multiplication processes. However, to make this point more clear we have added two references of INP measurements when discussing SIP.*

*The ice phase optical properties were found to have an insignificant effect on the radiative transfer in our simulations where the total extinction was kept fixed. Removing the ice phase would have likely led to increase in liquid phase extinction through increase in droplet number. Therefore our simulations should be considered as sensitivity study on the effect of the choice of the ice asymmetry parameter to radiative transfer. We have improved the discussion and interpretation of the results from the radiative transfer simulations to make this point more clear.*

*As suggested, we have improved the introduction by adding references to works, where the vertical structure of Arctic mixed-phase clouds was investigated, and added a "discussion and conclusions" section. The current conclusions section was renamed as summary section. More detailed description of the changes can be found below in our replies for the major and minor comments.*

Major/minor issues:

Line 22; specifically please see Gultepe et al BL clouds/ice fog work for Arctic clouds, this work is directly related to your work here.
*The following references to Gultepe et al. were added to the Introduction:*

*Gultepe, I., Isaac, G., Hudak, D., Nissen, R., & Strapp, J. W. (2000). Dynamical and microphysical characteristics of Arctic clouds during BASE. Journal of Climate, 13(7), 1225–1254.*

*Gultepe, I., Isaac, G. A., & Cober, S. G. (2001). Ice crystal number concentration versus temperature for climate studies. International Journal of Climatology, 21(10), 1281–1302.*

*Gultepe, I., A. J. Heymsfield, P. R. Field, and D. Axisa, 2017: Ice-Phase Precipitation. Meteor. Monogr., 58, 6.1–6.36.*

Ln35-40; see Ni-T for ice clouds based on various studies, and Gultepe et al showed that no trend in Ni-T (Intern J. of Climate)
*This reference was added. We also added a sentence summarising these observations:*
*"Gultepe et al. (2001) summarised ice crystal number concentration observations from two Arctic campaigns with maximum dimensions greater than 125 μm. The observed average number*

*concentrations measured over a wide temperature range from 0 and -45°C varied between 0.3 and 6.4 $L^{-1}$."*

Ln50; see Gultepe et al for SIP and ice crystal splintering issues (AMS met Monographs2018; Arctic ice cloud studies)
*We added this reference.*

Ln58; provide a ref for Ice fog/BL clouds high Ni values found in Arctic clouds, Lawson et al and Gultepe et al (Atmos Res review, 2017)
*We want to remind that in our literature study we summarise in-situ measurements covering only Arctic mixed-phase clouds, where the vertical profiles of low-level clouds were investigated. As mentioned in previous responses, we added reference to Gultepe et al. (2000) and Gultepe et al. (2001). We also added the reference Lawson et al. (2001), and added the following statement:*
*"On the contrary, Lawson et al. (2001) reported extremely high ice particle number concentrations (exceeding 1000 $L^{-1}$) for another cloud system measured during the same campaign observed at -12°C, and thus not explainable by rime-splintering."*

Ln68; why suddenly g became important here?
*The asymmetry parameter is a key single-scattering property for radiative transfer calculations but it is not necessary to highlight it in this part of the Introduction. We modified the sentence as following:*
*"We also discuss the the vertical variability of liquid and ice phase optical properties and discuss the implications of ice phase for the radiative properties of the low level clouds."*

Ln116; why cip not used for 15 micron bins? See Gultepe et al grey probe used for BL ice clouds and drizzle.
*We use CIP only for retrieving ice phase microphysical properties and for this, we only use particles with D>200 μm, due to uncertainties related to detecting asphericity and uncertainties related to sensitive volume of smaller particles.*

Ln152; why assumed as spheres? See Gultepe et al Ice fog review in Atmos Res 2017
*The SID-3 instrument measures the differential scattering cross section at the 50° scattering angle with a detector having a half-angle of 9.28°. This measured intensity is then calibrated using spherical particles so that the resulting diameter is the equivalent diameter of a sphere that would have the same differential scattering cross section. There is not an established way how to derive IWC from single particle light scattering measurements. Here we made the assumption to use the most simplified model for the ice crystal shape, i.e. a sphere, since the IWC in the measured mixed-phase clouds is mainly dominated by particles in the CIP measurement range (D>200 μm). The IWC in the SID-3 measurement range contributed typically less than 1% to the total IWC, so any uncertainty caused by the shape assumption would not significantly affect the total IWC.*

Eq. 4; what is Qeff? And what wavelength considered here? (See AMS Bull Arctic Ice Fog campaign)
*We are not sure what the Reviewer means by $Q_{eff}$. The Mie calculations to derive the scattering cross section were performed for the wavelength of 532 nm. The text was modified as following:*
*"The extinction coefficient for **visible** wavelengths was calculated using the following equation …*
*For spherical particles in the SID-3 size range, the $\sigma_{ext}(D)$ was calculated by multiplying the geometrical cross section with the extinction efficiency ($Q_{ext}$) calculated using the Mie theory for 532 nm."*

Fig. 2; where is RHw? If not saturated why we see SIP particles?
*The RH corresponds to RHw. This was made clearer in the figure caption. The measured RHw does show unrealistic low values. Therefore we also added a clarification to the figure caption:*
*"Note that the absolute value of relative humidity is not considered to be reliable and the values in panel B should only be considered to represent the trend in the relative humidity."*
*Since the reported cases are all mixed-phase clouds, we assume that the RHw = 100%, in which case RH_ice > 100%.*

Why not show RID?

*If the Reviewer means Rosemount icing detector, the instrument was not installed on P6.*

Same fig; you cant just plot wind speed like this, see Gultepe et al 1990 and 1995 Arctic cirrus studies.

*It is not clear what change to the presentation of the wind speed the Reviewer is suggesting. Fig. 2C shows statistical properties of vertical wind as a function of height. It is not informative to show the mean of the vertical wind as this fluctuates around 0. Variance gives information on the magnitude of the vertical wind fluctuations and thus give an indication of turbulence strength.*

Fig. 3; how did you filetered data to get solid lines???? From what? Aircraft?

*The solid lined show the average 10-s aircraft observations for each height bin. This is made now more clear in each figure caption.*

Fig 5; how did you removed the snow from the analysis? Snow versus cloud?

*We did not remove precipitation particles from the habit analysis. This is mentioned in the figure caption.*

Fig 6; same issue?

*How the solid lines were calculated was made clearer in the figure caption: "The mean and standard deviation were calculated based on 10-s aircraft observations for normalised altitude bins having bin edges at -0.6, -0.3, 0, 0.25, 0.45, 0.65, 0.85, 1, 1.05."*

Ln280; Gultepe et al (Atmos Res review, Arctic ice particles were not irregular), they showed it clearly also.

*This reference refers to Arctic ice fog studies of Gultepe et al. (2014) and Kim et al. (2014) and their findings of pristine crystals. We, however, refer to studies in mixed-phase clouds, which are more comparable to our results as the formation of ice crystals in ice fog and mixed-phase environments can be different.*

Fig 09; how did you interpret data? These are just selected ones, explain it.

*These are selected example crystals. The figure caption was revised as following:*
*"Examples of ice crystals representing three categories that were frequently observed on 2 June: large (D>500 µm) needles and columns, smaller unrimed faceted crystals and other polycrystals. Images are from the PHIPS probe."*

Fig 12; why you normalize the data??? Show T also, o idea what is T?

*The data is now shown as a function of actual altitude. The temperature at the cloud top is given in the text, which we consider to be sufficient, as the cloud systems were rather shallow.*

Fig 13; a has some relationship but not others, why?

*In the revised version, we are not discussing the trends in LWP, only in IWP.*

Section 6; not clear and results are not meaningful to me, please discuss and compare with others.

*Section 6 (now 5) was edited to improve its clarity. Discussion of our results in comparison with literature is not possible to our knowledge, as we are not aware of studies that investigate the sensitivity of radiative transfer in low-level mixed-phase clouds to ice crystal asymmetry parameter.*

Parag 425/conclusions; very well known statements, and I see that conclusions need to be improved and referenced for the knowledge given in the paper.

*We renamed the section as summary section. A detailed, referenced, discussion of the statements presented in summary can be found in the added discussion section.*

Finally, how did you discriminate SIP from the prime ice crystals? What is the basis?

*SIP was identified in the basis of knowledge of INP concentrations at T=-10°C, which are estimated to be well below $10^{-2}$ $L^{-1}$. Since the observed ice crystal concentrations were several order of magnitude larger than the INP estimate, we can assume with high confidence that the*

*observed ice crystals were the result of ice multiplication. This was made more clear in the discussion section:*
*"Previous observations in the Ny Ålesund have shown that springtime INP concentrations at -7°C are on average below $10^{-3}$ L$^{-1}$ with 95-percentile being below $10^{-2}$ L$^{-1}$ (Li et al., 2022). This is in consensus with the general knowledge of INP concentrations in this temperature range (Kanji et al., 2017). Therefore, it is highly likely that the observed ice crystal concentration in all of the presented warm and neutral period cases were results of ice multiplication."*

I suggest major revisions for this work, and see it again.